# Cumulative cultural evolution and mechanisms for cultural selection in wild bird songs

Heather Williams [1 ✉], Andrew Scharf [1,2], Anna R. Ryba[1,3], D. Ryan Norris [4], Daniel J. Mennill [5], Amy E. M. Newman[4], Stéphanie M. Doucet[5] & Julie C. Blackwood[2]

Cumulative cultural evolution, the accumulation of sequential changes within a single socially learned behaviour that results in improved function, is prominent in humans and has been documented in experimental studies of captive animals and managed wild populations. Here, we provide evidence that cumulative cultural evolution has occurred in the learned songs of Savannah sparrows. In a first step, "click trains" replaced "high note clusters" over a period of three decades. We use mathematical modelling to show that this replacement is consistent with the action of selection, rather than drift or frequency-dependent bias. Generations later, young birds elaborated the "click train" song form by adding more clicks. We show that the new songs with more clicks elicit stronger behavioural responses from both males and females. Therefore, we suggest that a combination of social learning, innovation, and sexual selection favoring a specific discrete trait was followed by directional sexual selection that resulted in naturally occurring cumulative cultural evolution in the songs of this wild animal population.

[1] Biology Department, Williams College, Williamstown 01267 MA, USA. [2] Mathematics and Statistics Department, Williams College, Williamstown 01267 MA, USA. [3] The Rockefeller University, 1230 York Ave., New York 10021 NY, USA. [4] Department of Integrative Biology, University of Guelph, Guelph N1G 2W1 ON, Canada. [5] Department of Integrative Biology, University of Windsor, Windsor N9B 3P4 ON, Canada. ✉email: hwilliams@williams.edu

When social learning by individuals results in population-level changes in a behavioural trait, the result is cultural evolution[1–3]. Observations of change over time in population-specific learned vocalizations[4–13] provide direct evidence for cultural evolution in wild animal populations[14]. Because social learning of vocalizations by songbirds has many parallels with the development of human speech[15–20] and those learned songs and calls play an important role in intra-specific communication[21,22], long-term field studies of the songs of wild bird populations are an excellent model system for studying cultural evolution in a natural context.

"Cumulative cultural evolution", which is especially prominent in humans, results when successive rounds of cultural evolution refine a learned behaviour[23,24], producing a ratcheted series of improvements[25–27]. The "core criteria"[25,28] for demonstrating cumulative cultural evolution are: i) a change in a learned behaviour, that is ii) transmitted via social learning to other individuals, where iii) the new behaviour results in an improvement in performance or "efficacy", followed by iv) a later repetition of steps i-iii that results in additional increments of change in the same behaviour. In non-human animals, direct evidence for cumulative cultural evolution comes from managed[29] or captive populations[30]. Examples include the regeneration of species-specific characteristics in domesticated zebra finch songs[31] *(Taeniopygia guttata)* and the adjustment of routes by homing pigeons[32] *(Columba livia domestica)*. Indirect or incomplete evidence suggests that cumulative cultural evolution has played a role in the tool use of wild populations of birds[33] and primates[30,34], the feeding behaviours of Japanese macaques[35] *(Macaca muscata)*, and the songs of humpback whales[27] *(Megaptera novaeangliae)*. However, direct evidence that satisfies all four of the core criteria for cumulative cultural evolution in naturally-occurring behaviours of a wild population is lacking.

We previously described the replacement of one song characteristic by a novel form that resulted in greater reproductive success in a wild population of Savannah sparrows[36] *(Passerculus sandwichensis)*. Here we describe a second round of cultural evolution in the same song trait: (i) a new form variation in the trait, (ii) social learning of the new variants by later generations, (iii) resulting in increased efficacy of the song. This repeated round of changes in the same song trait satisfies the fourth core criterion for cumulative cultural evolution and provides a fully documented example of naturally-occurring cumulative cultural evolution in a wild population.

We also ask which mechanisms could have been responsible for the two rounds of cultural evolution that we observed in Savannah sparrows' songs. Variation in a learned trait may result from copying errors, immigration of individuals with a different form of the trait, or innovation/improvisation during learning. The new variant of a behavioural trait may then change in prevalence either due to the random processes of cultural drift[37–39], or because of cultural selection[40]. Frequency-dependent learning biases represent one type of cultural selection. When a common behavioural form is preferentially learned, a conformist bias exists[41], while a rare-form bias results in the preferential learning of novel behavioural features[42]. In contrast, what we will call simply "selection" and some others call "direct selection"[2,43] shapes cultural evolution when social learning is guided by individuals' observations of the acoustic characteristics and social environment associated with a particular behaviour. Such selection can result from "prestige bias"[44] – based on the characteristics of the individuals performing the behaviour[45] (e.g. copying a dominant or successful individual's song), or "payoff bias"[46] – based on observation of the outcomes of different behavioural variants (e.g. copying songs with acoustic characteristics that result in improved transmission through the environment[47]).

Selection can also be based on sensory predispositions[31] that make specific acoustic characteristics of a song more attractive to learners. A sexual component of selection is likely to be important for the cultural evolution of learned birdsongs, which are used in defending territories and attracting mates.

In this study, we assess the relative importance of potential mechanisms for the cultural evolution of songs that we observed by modeling the social learning of song based on data from multi-year field observations of demographics and behaviour. We then use our model to assess how well the observed pattern of multigenerational changes in a learned song feature is predicted by three different mechanisms: 1) selection (which should result in a steady increase in the prevalence of a new trait), 2) drift (characterized by random fluctuations in trait prevalence) and 3) frequency-based learning bias (the common variant favored by conformity, or an equilibrium between traits in the case of rare-form advantage). We also hypothesize that sexual selection could be an important component of cultural selection on song because many previous studies have demonstrated that song is important for mate choice in birds[21]. To assess the relative importance of the genetic fitness of the singer and the cultural fitness of the song, we compare males' survival rates, to the transmission rates of their songs. Finally, we ask whether different forms of selection might be responsible for successive incremental changes in the cumulative cultural evolution of song features.

## Results

**Replacement of a learned song feature.** Savannah sparrows *(Passerculus sandwichensis)* are small (18 g) migratory songbirds that breed in North American grasslands[48]. Nearly all male Savannah sparrows crystallize one song during their first year, which they then sing for the rest of their lives (females do not sing)[49]. We recorded songs of individually identified birds from a highly philopatric population breeding on Kent Island (New Brunswick, Canada; Fig. 1)[50–54] in 1980, 1982, 1993–1998, and then continuously from 2003 to 2019[49].

Songs have a consistent, four-part structure (Fig. 2a), and different segments of the song change at different rates, with a buzz segment that remains consistent within a population[55] and a middle portion that varies considerably within a population[36]. In this study we focus on the song's "interstitial notes", a term we apply to the soft notes sung in the intervals between successive loud introductory notes. During the 35 years of our long-term study, there were two main forms of these interstitial notes: "high note clusters" (Fig. 2b) are a sequence that usually includes three distinct note types, while "click trains" (Fig. 2e) include only one note type, a repeated short click (see Supplementary Fig. 1 for a full description of note types). We have shown previously that high note clusters began to be replaced by click trains between 1983 and 1987, that both forms appeared in songs recorded between 1988 and 2009 (Fig. 2c), and that males singing click trains had nests that produced more fledglings in 2002-4[36]. The replacement of high note clusters by click trains was complete by 2010 (Fig. 2e).

A second step in the cultural evolution of this song feature began in 2004, after click trains were well established and were sung by more than 76% of the population (this breakpoint was determined by segmentation analysis, $t = 5.83$, $p < 0.0001$). Prior to 2004, click trains included 2–5 clicks between introductory notes (Fig. 3a), and the average number of clicks (2.9) did not change between 1993 and 2003 ($R^2 = 0.006$, $F_{1,153} = 1.0$, $p = 0.32$). From 2004 onwards, the mean number of clicks sung per train increased across years (Fig. 4a, $R^2 = 0.18$, $F_{1,455} = 98.6$, $p < 0.0001$), and was correlated with the proportion of males singing click trains in their songs that year (Fig. 3b, $R^2 = 0.87$,

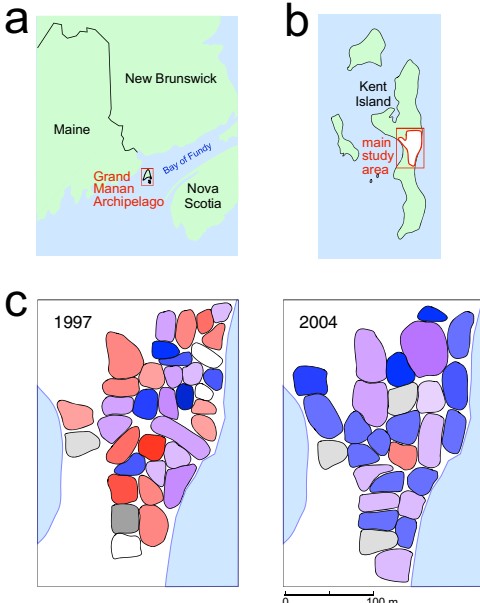

**Fig. 1 Study site location and distribution of click train and high note cluster singers' territories. a** The Grand Manan Archipelago lies in the southwest Bay of Fundy; Kent Island is the southeasternmost island in the archipelago. **b** Kent Island and the location of the main study site, which includes the largest contiguous set of territories that were followed continuously (images in a and b modified from Google Maps). **c** Representative examples of territories within the main area of the study site. In 1997 click trains and high note clusters were equally represented, and in 2004 click trains were sung by the majority of males. Blue = territories of males singing click trains, red = males singing high note clusters, and purple = males singing both features. Territories of older birds are shown with darker shades. Territories of males whose songs were not recorded are shown in white and those with songs that included neither introductory feature are shown in gray. Depending upon conditions, songs can be heard from 50 to 150 m from the singer.

$n = 10$, $p < 0.0001$). At the same time, variation in the number of clicks in a train also increased as more males sang click trains (Fig. 3c; $R^2 = 0.62$, $n = 17$, $p < 0.001$). These increases in the range, average, and population-wide variation in the number of clicks sung in a male's train began more than 15 years (and generations) after click trains were first recorded.

The progressive increase in the number of clicks that started in 2004 could have occurred in one of two ways: (1) older birds added clicks to their songs from year to year (individual change), or (2) younger birds sang more clicks than were present, on average, in the songs sung by males they copied (generational change). Between 2004 and 2013 there were 230 cases of birds returning to breed after their first year; only 8 (3.5%) changed the number of clicks in their songs between years (four increases and four decreases). The number of clicks in the songs of first-year breeding males averaged 0.27 more than in those of older birds present in the same year ($F_{(1,454)} = 8.34$, $p < 0.005$; Supplementary Fig. 2). We conclude that the increase in number of clicks is primarily due to first-year males incorporating more of them into their songs during learning. This conclusion is reinforced by the observation that the first recordings of songs including 6, 7, or 8 clicks were all from first-year breeders.

**Playback study**. To determine whether birds responded differently to click trains of different lengths, we conducted a playback experiment in 2011. Each of 25 male playback subjects was pre-sented with four introductory segments of songs that differed only in

the length of their click trains (0, 2, 4, or 7 clicks; see Methods). This range of clicks corresponded to those in the 39 songs recorded on the study site at the time of the playback study, with 4 clicks being the most common form ($n = 16$); trains with 7 clicks and 0 clicks were equally rare ($n = 3$). These auditory stimuli evoked species-typical aggressive responses: males flattened their feathers, crouched low, flew or ran towards the sound, and fluttered their wings aggressively[48]. Stimuli with more clicks in each train elicited responses with longer durations (Fig. 4d; $F_{(1,73)} = 10.97$, $p < 0.005$). In 11 of the playback presentations, females also responded to the stimuli, but not with aggressive behaviours: instead they stood erect, raised their head feathers to form a small crest, and hopped towards the speaker, looking around as if to locate the source of the sound. Females' first approach to the speaker occurred disproportionately more often when the stimulus included a train with 7 clicks (Fig. 3e; $X^2 = 11.69$, df = 3, $p = 0.009$). The nature of males' and females' stronger responses to longer click trains suggests that more clicks make a train more effective – in terms of both male competition and female choice.

**Modeling mechanisms for cultural evolution**. To investigate the evolutionary mechanisms that resulted in the replacement of high note clusters by click trains, we used a discrete time dynamical model[56] to describe how the songs in this Savannah sparrow population would change as a result of (i) drift, (ii) frequency-dependent bias, and (iii) selection. The model incorporated features of the birds' life history, demographics, and song learning based on long-term data from Kent Island[57] (for details see the Methods). Although spatial patterns can be important for the dynamics of language loss[58], territories with birds singing click trains and high note clusters were intermixed and no spatial structure was apparent (Fig. 1), so we did not include spatial distribution in the model. We used information derived from song recordings and guidance from the literature to set initial model parameters: two innovators (2.9% of the study population), first appearing in 1983, singing both high note clusters and click trains as a blended trait (see the Methods for the rationale for these choices). We later tested the effect of altering our choices of values for the initial parameters (see below).

We compared the model's predictions to observations of songs over 35 generations between 1980 and 2013. To evaluate the relative importance of frequency-dependent learning biases (β) and selection (σ) in song learning by first-year birds we used a Type III Holling response curve[59]. We calculated maximum likelihood estimates (MLEs) to test how well model outcomes fit the long-term data for the following four cases: (1) cultural drift (no learning bias and no selection, β = 1 and σ = 1); (2) frequency-dependent bias in the absence of selection (σ = 1 and varying β); (3) selection in the absence of frequency-dependent bias (β = 1 and varying σ); and (4) a combination of frequency-dependent bias and selection (varying both β and σ).

The "cultural drift" or neutral model, did not include either frequency-dependent bias or selection (values for both β and σ were set to 1). This model did not produce results that matched the historical data (Fig. 4b; ΔAIC = 82.0; Supplementary Table 1). Instead, click trains either disappeared altogether or persisted only in a small proportion of males' songs.

We next considered the role of frequency-dependent learning operating alone, setting selection to be neutral (σ = 1) and varying the frequency-dependent bias parameter β from 0.5 (a strong rare-form bias) to 2 (a strong common-form bias). The version of the frequency-dependent bias model that best fit the data had a moderate rare form bias (β = 0.74) and a poor fit to the historical data (ΔAIC = 67.6; Supplementary Table 1). This model resulted in a consistent and stable outcome: one-fourth of

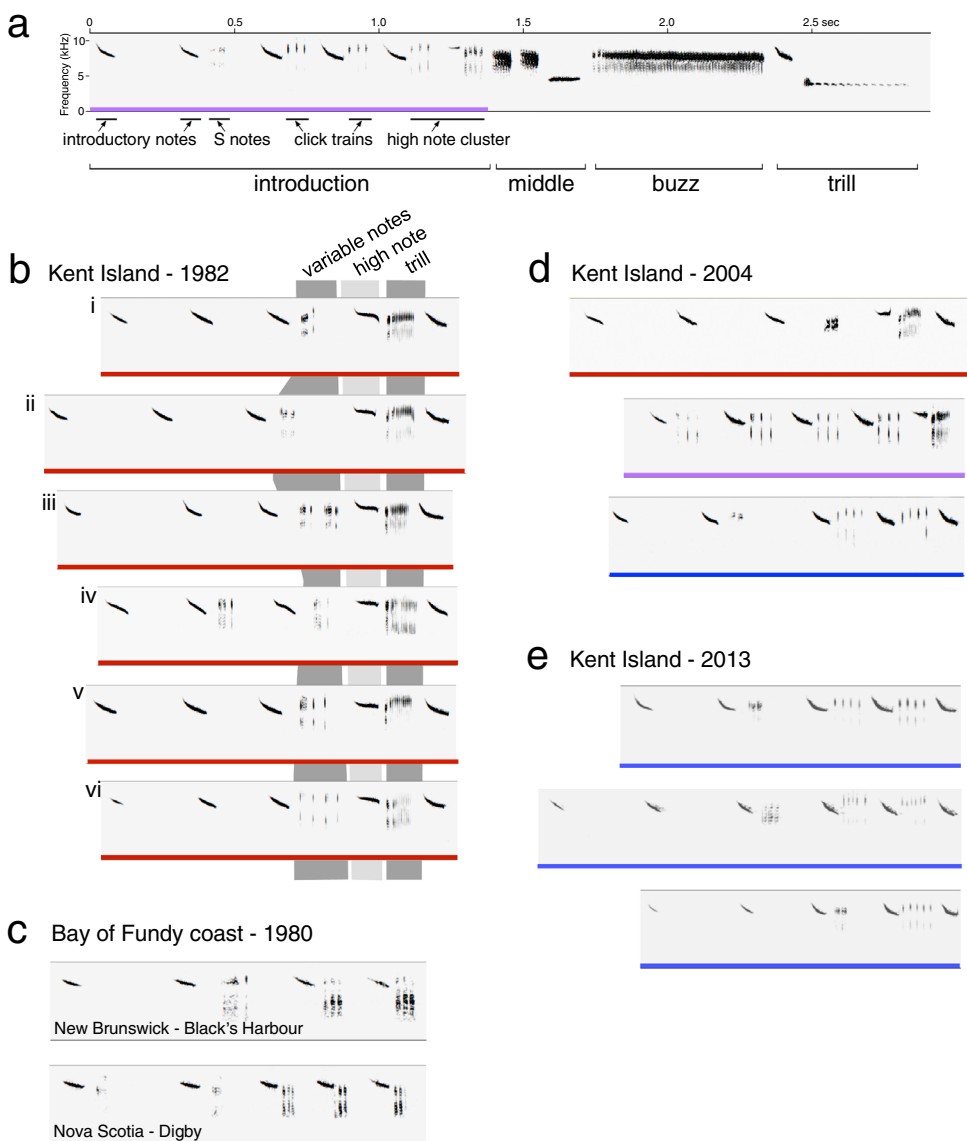

**Fig. 2 Sound spectrograms of Savannah sparrow song and introductory segment features. a** Savannah sparrow song, showing all segments. The introductory segment includes softer interstitial notes after the later loud introductory notes: two click trains (sequences of identical short click notes) as well as a high note cluster with three different note types. **b** Introductory segments recorded on Kent Island in 1982, showing the three sections of their high note clusters (variable notes, high note, and trill). (i) The most common form (19 of 42 males recorded) with an S note and a click as the variable notes (see Supplementary Fig. 1 for a description of types of variable notes). (ii) The second most common form (n = 4) included the same variable notes. Forms iii–vi were each sung by a single male. (iv) This "stuttered" form duplicated the first part of the high note cluster in the penultimate interval between introductory notes. (vi) In this song the variable note portion consisted solely of clicks; these do not form a click train because other note types are also sung between the two introductory notes. No click trains occurred in any of the songs recorded on Kent Island in 1980 and 1982. **c** Two representative songs from recordings of nearby mainland populations in 1980, including triplets of a different interstitial note type (see Supplementary Fig. 1 for the differences between clicks and these "X" notes). **d** The three introductory segment types sung on the study site in 2004. Colour coding: red = high note cluster, blue = click train, purple = both features. **e** Representative introductory segments from 2013, including click trains with 4, 5, and 6 clicks.

the males sang click trains, one-fourth sang high note clusters, and half the population sang both forms (Fig. 4c), which did not match the replacement that actually occurred. The failure of frequency-dependent bias to match the observed data is not surprising, because a common-form learning bias would stabilize an existing song form and prevent novel cultural traits from increasing in frequency, while a rare-form bias results in the rarest variant increasing in frequency until it becomes common – at which point it is no longer favored. Thus frequency-dependent bias alone cannot account for the replacement of high note clusters by click trains.

We then modeled the effect of selection alone on the prevalence of click trains and high note clusters in the absence of frequency-dependent bias by setting β to 1 (neutral), and varying the selection parameter σ from 0.5 (strong selection against click trains) to 2.0 (strong selection for click trains). The best-fitting version of the selection model featured moderate to strong positive selection (σ = 1.70) favoring click trains, and achieved a good fit to the historical data (Fig. 4d; ΔAIC = 0; Supplementary Table 1), with an initial increase in "mixed" songs including both high note clusters and click trains followed by the loss of high note clusters and fixation of click trains.

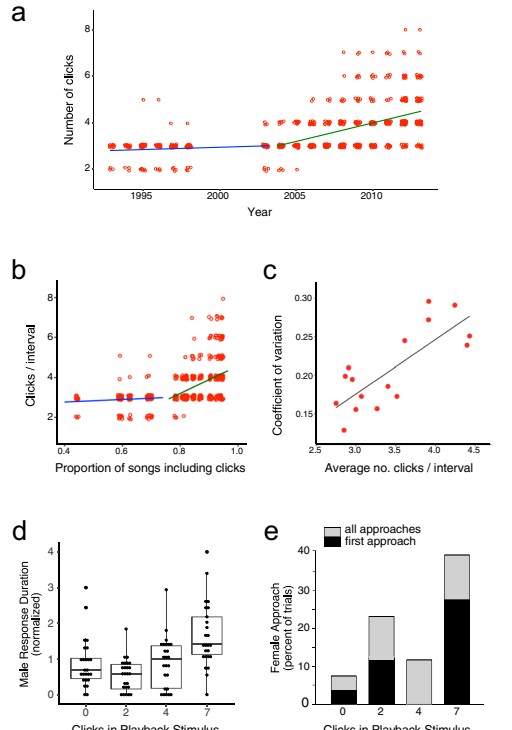

**Fig. 3 Cumulative changes to click trains and responses to playbacks of click trains. a** The number of clicks in a train was stable until 2003 and increased thereafter. **b** The number of clicks in a train as a function of the proportion of the population's songs that included click trains. The breakpoint (75%) corresponds to 2004; see text for details. **c** As the number of clicks in a train increased, so did the coefficient of variation. **d** Click trains with 7 clicks elicited longer-lasting aggressive responses from males ($n = 25$) in playback experiments (the centre bar shows the median, boxes the 25th and 75th percentiles, and the whiskers the 95th percentile). **e** Females that responded ($n = 11$) were more likely to approach click trains with 7 clicks (black portion of bars = female's first approach to the playback speaker; gray bars = approaches to the speaker in subsequent trials). Source data are provided in the Source Data file.

Finally, to determine whether frequency-dependent bias and selection work together to account for the replacement of high note clusters by click trains, we varied both the frequency-dependent bias (β) and the selection (σ) parameters from 0.5 to 2.5. The results of this "full model" were essentially identical to those of the "selection only" model; the two models had the same AIC values and nearly identical values for the parameters (Fig. 4e; ΔAIC = 0; Supplementary Table 1). In the full model, moderate to strong selection (σ = 1.71) favored click trains, and there was effectively no frequency-dependent bias (β = 0.99 ≈ 1). The absence of a role for frequency-dependent bias in the full model highlights the importance of selection in the replacement of high note clusters by click trains in the Savannah sparrows' songs.

We then examined some of our model's assumptions. We first asked whether songs with both click trains and high note clusters are best represented as a single blended trait (half click train and half high note cluster) or as including two different traits. The model that treated the presence of both features in a song as a single blended trait better fit the historical data (Supplementary Table 2 and Supplementary Fig. 3), validating our use of the blended trait in the main model.

Next we asked how the model's results were affected by changing the year in which click trains were introduced and the number of innovators (first-year birds introducing the click trains

into the population). We varied the introduction of click trains from 1983 to 1987, the range of possibilities defined by the recording data. Earlier introduction yielded the best fit to the historical data, but differences in the model's results across years were relatively small (see Supplementary Table 3 and Supplementary Fig. 4). Finally, we varied the number of innovators from 1 to 8 (a mutation rate ranging from 0.014 to 0.114). Although including larger numbers of innovators in the model did produce a better fit to the data, the values for frequency-dependent learning bias and selection were similar across this wide range of innovators (see Supplementary Table 4 and Supplementary Fig. 5). Thus, varying the number of innovators and the timing of introducing the innovation did not change the model's primary result: selection alone, with no contribution from a frequency-dependent learning bias, accounted for the replacement of high note clusters by click trains.

**Source of the new song form**. We also considered the question of whether click trains first arose because of a) immigration of individuals that learned the form elsewhere or b) innovation or improvisation based on existing local song forms. We looked for potential sources of immigrants singing click trains among songs recorded in 1980 from 11 island and mainland locations close to Kent Island as well as in 74 archived recordings drawn from 32 locations in northeastern North America over several decades[55]. High note clusters occurred in the songs of four populations near Kent Island (on two islands in the Bay of Fundy and on the adjacent coasts of New Brunswick and Nova Scotia). Although birds in some other nearby populations sang triplets of longer interstitial notes (5–8 ms X notes, with two amplitude peaks; see Supplementary Fig. 1) between their introductory notes (Fig. 2c), none sang trains of clicks, which are shorter (2–3 ms), and have a single amplitude peak. We also looked for evidence of innovation or improvisation based on existing Kent Island notes. Of the forty recorded 1982 Kent Island songs, one included four clicks as the first segment of variable notes within the high note cluster (Fig. 2b, vi), and another included two clicks as one of the note types within the first segment of the high note cluster (see Fig. 2b, v). A third bird "stuttered" and sang the first portion of his high note cluster (which did not include clicks) in the interval between introductory notes that immediately preceded the high note cluster itself (Fig. 2b, iv). If a bird with a high note cluster that began with clicks stuttered, singing the variable note segment before delivering the full cluster, the resulting song would have had a click train and a high note cluster (separated by an introductory note) – the form of the first click train song to appear in our recordings. Such an innovation appears to be the most likely source of click trains.

**Cultural selection acts on the song, not the singer**. Once click trains were present in the population, our modeling suggests that moderate-to-strong selection (σ ≈ 1.7) was responsible for their rapid increase within the population. Why might click trains have been favored? One possibility is that the adult males that learned to sing click trains had some inherent fitness or developmental advantage[60], which would then be reflected in higher survival rates. To assess this possibility, we compared the survival of adult males singing click trains to those singing high note clusters in the years 1994–1998 and 2004–2008 (years in which both forms were present and for which we have comprehensive song recordings in two subsequent years). The survival rates of adult males that sang high note clusters (w = 1.01) did not differ from those of adult males that sang click trains in the same year (w = 0.98; paired t = 0.45, df = 9, p > 0.66). In contrast, when we compared the transmission via copying of these two features in the songs of first-year breeding males relative to the adult population they copied (the adult song models), click trains

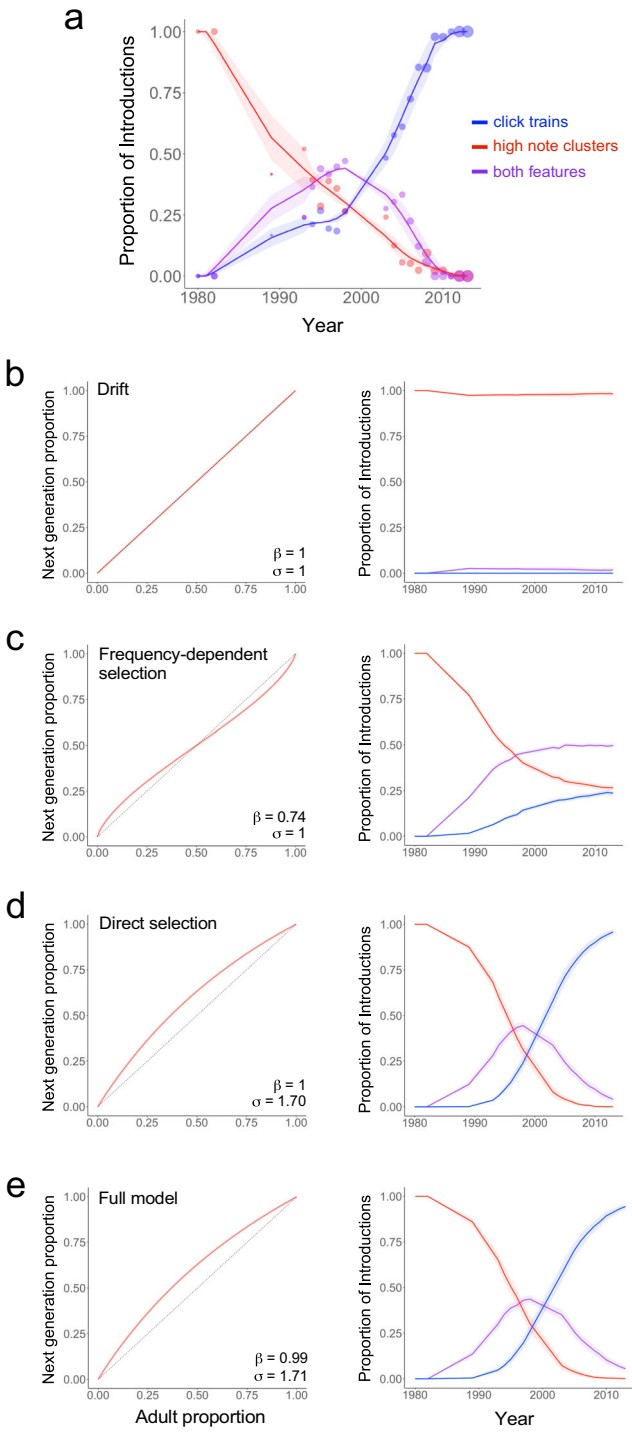

**Fig. 4 Modeling replacement of high note clusters by click trains.**
**a** Historical data showing proportions of songs with high note clusters (red), both high note clusters and click trains (purple), and click trains alone (blue). Point sizes are proportional to numbers of songs recorded in a given year, solid lines are splines fitted to each category, and shading represents the 95% confidence interval. **b**–**e** The learning curves (left) and outputs of model (right, with 95% confidence interval error bands) and the frequency-dependent bias and/or selection parameters that generated the best fit to the historical data for each type of model. **b** Cultural drift model. **c** Best-fitting frequency-dependent learning bias model ($\beta = 0.74$ represents a moderate rare-form bias). **d** Best-fitting selection model ($\sigma = 1.70$ represents moderate to strong selection for click trains). **e** Best-fitting full model that simultaneously varied selection and frequency-dependent bias. The best version of this model is nearly identical to (**d**) with essentially no frequency-dependent bias ($\beta = 0.99 \approx 1.0 =$ no frequency-dependent learning) and moderate to strong selection ($\sigma = 1.71$). Source data are provided in the Source Data file.

had previously demonstrated that the replacement of high note clusters by click trains in the songs of Savannah sparrows was an innovation that was learned by new generations, and that singers of click trains produced more fledglings. When a subsequent change in the same behavior also completes the cycle of variation, social learning, and improved efficacy, the incremental changes represent cumulative cultural evolution. Here we have shown that, after they had been adopted, click trains began to vary in length, that longer click trains were preferentially learned, and that these longer click trains elicited increased aggression from males and interest from females. This second round of cultural evolution of the same behavioral trait satisfies the fourth core criterion for cumulative cultural evolution[27]. Savannah sparrow song thus provides a fully documented example of naturally occurring cumulative cultural evolution in an unmanipulated wild animal population.

It will be interesting to follow this Savannah sparrow population, both to study further steps in the cultural evolution of song and because our data suggest interesting parallels to other examples of culturally evolving vocalizations. In humpback whale songs, one song form is regularly replaced with another[61]; corn buntings incrementally change song notes from year to year[62]; and a white-throated sparrow song form has been spreading across North America during the past two decades[13]. The differences in time and spatial scales across these examples need not preclude the existence of similar patterns of cultural evolution. Humpback whale songs are notable for a long-term cycle of small evolutionary changes followed by a "revolutionary" replacement of the existing shared song type with a new, less complex song[5]. The Savannah sparrow high note cluster, which included at least three note types, was replaced by the less complex click train, which has only one note type. The subsequent increase in click number may represent an increase in complexity. Similar mechanisms may be responsible for these parallels in the cultural evolution of different species' vocal communication systems.

Variation in the songs of Savannah sparrows most likely arose during song learning. Although we did not directly observe the introduction of click trains, we considered three possibilities: (1) immigration, (2) first-year males learning click trains on the wintering grounds, and (3) innovation. Neither recordings of songs from other populations nor unusual songs recorded on Kent Island (see Supplementary Fig. 7) provided evidence to support the idea that immigrants introduced click trains. The second explanation, that young males may have heard and learned click trains on the wintering grounds, is an intriguing possibility; Kent Island Savannah sparrows overwinter in a variety

($w = 1.13$) had a significantly higher transmission rate than high note clusters ($w = 0.77$; paired $t = 2.44$, df = 9, $p < 0.05$; Supplementary Fig. 5). Although males with high note clusters and click trains were equally likely to return in subsequent years, young males were more likely to copy click trains than would be predicted by the proportions of each form they heard during their hatching year.

## Discussion
Cultural evolution occurs when variation in a behaviour is followed by social learning of the new behaviour, and that new form of the behaviour results in improved performance, or efficacy. We

of locations[63], potentially giving each young male a different set of options for winter learning. However, our recordings of the crystallized songs of returning first-year males banded as nestlings or fledglings on Kent Island do not include song elements that are foreign to the population. Furthermore, Savannah sparrows may not sing on their wintering grounds[48]. Thus we favor the third explanation, that of innovation (or copying errors) based on existing songs. The developmental innovation explanation is supported by our observation of how clicks were later added to click trains. Early in the season, the number of clicks sung in a first-year male's click train varied by as many as two clicks within a song bout and often included more clicks than were present later in the stable crystallized song. Thus we suggest that variation in the songs arises late during song learning, perhaps even after first-year males return to the breeding area in the spring. At that time they routinely sing more than one plastic song type and then crystallize one form[64,65] (as do other songbird species[66]). Experimentation and innovation during the plastic phase of song learning[67,68] allows young birds to extend the range of their song characteristics during learning and so can result in rapid change, as occurred after 2004 with the increasing number of clicks in trains. The developmental innovation mechanism for generating variation is simple, and does not rely on the introduction from elsewhere of a novel form that is absent in song recordings from other populations.

To assess which mechanisms might have favored the social learning of a novel trait (click trains) rather than the trait prevalent within the population (high note clusters), we modelled the effects of cultural drift, frequency-based bias, and selection. The model's results strongly suggest that cultural selection, rather than cultural drift or frequency-dependent learning biases, best explains the spread of the new song feature through the population. Previous modeling studies of vocal dialects suggest that population-wide stability is maintained by a conformist learning bias[41,69], and that cultural drift is responsible for variability in songs over time[37]. In contrast, we find little support for drift or frequency-dependent learning bias and strong support for cultural selection that directly favored click trains over high note clusters. The S-shaped trajectory that describes the increase in the prevalence of click trains within the population is a characteristic of cultural selection[70], and is also seen in replacements of human cultural variants by competing forms[71,72] and in changes in human language[73,74]. While conformist biases can stabilize song features over time, cultural selection may also result in apparent conformity because a song trait that has a selective advantage spreads through the entire population via social learning.

The mode of cultural selection differed between the two rounds of cultural evolution of Savannah sparrow interstitial notes. Initially, cultural selection favoring one of two different discrete traits led to the replacement of high note clusters by click trains. The second round of cultural evolution, which resulted in an increase in click numbers, is reminiscent of classic examples of directional selection on a continuous trait[75]. However, in the case of click train length, which varied because of innovation during social learning, cultural selection resulted in increased variation of the trait, in contrast to the reduction of standing variation that occurs when a heritable trait is winnowed by directional selection[76]. Different mechanisms and different forms of selection operating in succession to reshape the same socially learned trait may be a general feature of cumulative cultural evolution.

Both the replacement of high note clusters by click trains and the increase in number of clicks within a train resulted in increased efficacy of the song. During years when the two forms were equally common, males singing click trains fledged more offspring than those singing high note clusters[36]. As we have shown here, longer click trains elicited stronger responses by both males and females. Both results imply an important role for sexual selection in the context of cultural selection. Since extra-pair copulation is common in Savannah sparrows[77], a song that provides an advantage in attracting females and deterring other males is likely to be important in terms of male reproductive success. Small territory sizes[48] provide many opportunities for females to compare and respond to songs and for males to observe the outcomes of such interactions. It is likely that some combination of 1) demonstrator or payoff bias and 2) female sensory predispositions[2,78] (which may themselves be learned) is responsible for sexual selection on, initially, the learning of click trains and also for the subsequent round of cultural evolution that increased the number of clicks in a train. The cumulative cultural evolution we observed in Savannah sparrow songs is thus more akin to that of human social artefacts such as language[79], pottery ornamentation styles[80] or music[81] than to that of human material technology[82]. In these realms, the distinction between "functional" and "stylistic" changes is often tied to mechanisms: stylistic changes are due to drift, while functional changes are due to selection[83]. We have shown that drift cannot account for the changes in Savannah sparrow song interstitial notes, which are due to selection, specifically sexual selection. That this selection is due to preferences that may be based on sensory predispositions or may themselves be learned (and evolve) makes the scenario more complex and more interesting.

Our data also show that the shift to longer click trains was based on selective copying and innovation by recruits to the population rather than being correlated to survival fitness of adult singers. Because songs can be learned from any of several adult models a young male hears[49], a male does not necessarily pass his song on to his offspring, even if singing that song has conferred a reproductive advantage. As a result, reproductive fitness (measured as the number of offspring a male fathers) need not be coupled to cultural fitness (measured as the number of individuals that copy a male's song). Since genetic traits related to survival were transmitted independently of socially learned songs, cultural selection acted on the properties of the song a male sang rather than on characteristics of the singer himself.

Innovation coupled with cultural selection causes changes in socially learned behaviours that are mediated by social interactions, as our data suggest for the changes we observed in click trains. Traits acquired through social learning have higher mutation rates as well as modes of transmission that can be independent of genetic relationships[49], and these differences yield faster rates of cultural evolution and differentiation compared to genetic traits[84]. Innovation and social learning provide an escape from the constraint of producing only traits already present in the population, as individuals can also improvise upon and thus extend the traits they learn beyond the range of the traits they copied. The consequent increase in variation may be a signature of directional cultural selection. When coupled with directional selection, innovation and social learning provide a powerful mechanism for accelerating cultural evolution. The relatively rapid, step-wise evolutionary changes in a learned vocalization that increased the efficacy of Savannah sparrow song represent spontaneous, naturally occurring cumulative cultural evolution in a wild animal population. Although what we describe here is simpler than cumulative cultural evolution in humans, this result adds to the parallels between bird song and human language. Cumulative cultural evolution may prove to be a general phenomenon in socially learned animal behaviours.

## Methods

**Study population and song recordings**. All animal procedures were carefully reviewed by the Williams College IACUC (WH-D), the Bowdoin College Research

and Oversight Committee (2009–18), and the University of Guelph Animal Care Committee (08R601), and were carried out as specified by the Canadian Wildlife Service (banding permit 10789D).

We studied Savannah sparrows (*Passerculus sandwichensis*) at the Bowdoin Scientific Station on Kent Island, New Brunswick, Canada (44.5818°N, 66.7547°W). Since 1988, individuals nesting within a 10 ha study area in the middle of the island (30–70 pairs each year; part of a larger population of 350–500 males breeding on Kent Island and two adjacent islands) have been colour-banded to facilitate visual identification, and complete demographic information is available for birds on the study site (though not for the entire population) for the years 1989–2004 and 2009–2013. Because of strong natal and breeding philopatry[51], birds hatched on the study site itself represent 40–80% of adult breeders in that area, and because of the systematic banding program, ages are known. Each year adds a new generation to the population, with yearlings making up approximately half of the adult breeding males. The birds banded and recorded on the study site are estimated to make up 10–20% of the Savannah sparrow population on Kent Island and two nearby islands.

Details of the recording methods used in this study (covering the years 1980, 1982, 1988-9, 1993-8, and 2003–13) can be found elsewhere[36,49]. Using digitally generated sound spectrograms (using SoundEdit Pro and Audacity), birds were scored as having either a) high note cluster=a final introductory segment interval including at least two different note types, or b) a click train=one or more introductory segment intervals including at least two clicks and no other note types, or c) both features[36] (see Supplementary Fig. 1 for a full description of note types). Although a small proportion of birds (mean = 8.3%) did not include either feature in their songs (such birds either had no feature in the introductory segment intervals or one non-click note type in the final interval), we did not include this option in the model and omitted these birds from summaries of the data. We did not include data after the breeding year 2013 because of we began an experimental field tutoring study in the summer of 2013[64].

**Modelling**. We used a dynamic, discrete time model which allowed us to focus our analysis to specific time points within the year that are related to song learning (the beginning and end of the breeding season). These were: (1) the return of older birds between breeding seasons, (2) the recruitment of young birds singing newly crystallized songs in the spring, and (3) reproduction, resulting in the addition of juveniles during the summer breeding season.

Because survival data were not available for every year during the time span we studied, we captured the variation in survival rates observed in the field[57] by using a binomial distribution centered on the average historical survival rate for each age class (addressing the possibility that cultural drift resulting from random differences in survival rates was responsible for the shift in song features). The model incorporates stochasticity to capture the variation in population dynamics and return rates by assigning parameter values for survival and return rates from empirically generated probability distributions.

We did not include spatial distribution of song variants in the model; although spatial patterns can be important for the dynamics of language loss[58], territories with birds singing click trains and high note clusters were intermixed and no spatial structure was apparent (Fig. 3).

The model assumes that males choose which features to incorporate into the introductory sections of their songs during song development. Individuals fall into

one of six mutually exclusive classes of male Savannah sparrows. The classes are defined by (1) the bird's developmental stage in the song learning process: juvenile (J, the first year, when the song is plastic) or adult (A, after the first spring, when the song is crystallized), and (2) the variant or variants sung as part of the bird's introduction (high note clusters, click trains, or both). Denoting note high note clusters with X and click trains with C, the adult classes are therefore AX, AC, and AXC, and the juvenile classes are JX, JC, and JXC. The sum of the individuals in these classes is the total male population.

We used two times during each year – late spring and late summer – to correspond to stages in song development (Fig. 5). At a given time $t$, when breeding is underway in the late spring, the male population consists entirely of adults singing crystallized song, and therefore each juvenile class is empty. At the end of the summer, the population of males has been augmented by juveniles, which are initially assigned to the same variant class as their fathers. To capture these dynamics, we define an intermediate time step, denoted $t^i$. Time $t + 1$ then corresponds to the following breeding season (late spring), when juvenile males hatched the previous year have completed song development, crystallized their songs, and joined the adult class.

In the late summer the male population increases with the addition of juveniles hatched that year, some of which will return to join the singing population the following year; survivors will return to breed within a few hundred meters of where they hatched[51]. To fit the observed historical decline in the Kent Island population[57], the total number of returning juveniles, r (including both those hatched on site and those immigrating from nearby populations at time), follows a Poisson distribution where m = 33.6 – .182x and x is the number of years since 1980 (this function results in a decline of 5 males per decade; the initial number on the study site used in the model, 70, was extrapolated from historical data). The size of each returning juvenile class at time $t^i$ then takes the form:

$$JY_{t^i} \sim \text{Poisson}(m) \frac{AY_t}{AX_t + AC_t + AXC_t} \quad (1)$$

for each Y ∈ {X, C, XC}.

After the following winter, the proportion of surviving adults at time $t + 1$ follows a binomial distribution where the mean survival rate s = 0.48 is derived from historical data. Therefore, each adult class takes the form:

$$AY_{t+1} \sim \text{Binomial}(AY, s) * AY_t \quad (2)$$

At the beginning of the next breeding season, juveniles complete song learning[64], choosing which variant to crystallize as part of the song, and enter an adult song class; thus all of the juvenile classes disappear at $t + 1$. Which adult class juveniles join depends on separate learning functions for each of the two variants, $\phi_X$ for the high note cluster and $\phi_C$ for the click train. The $\phi$ function takes values between 0 and 1 and gives the probability of crystallizing a song form during the transition from natal year to breeding, depending upon the frequency-dependent bias and selection parameters (see below). These functions define the proportion of features that appear in the next generation as compared to that of the previous generation. Therefore we have:

$$AX_{t+1} = (\phi_X)^2 JX_{t_i} + (1 - \phi_C)^2 JC_{t_i} + \phi_X(1 - \phi_C)JXC_{t_i} + AX_{t_i} \quad (3)$$

$$AC_{t+1} = (1 - \phi_X)^2 JX_{t_i} + (\phi_C)^2 JC_{t_i} + (1 - \phi_X)\phi_C JXC_{t_i} + AC_{t_i} \quad (4)$$

$$AXC_{t+1} = 2\phi_X(1 - \phi_X)JX_{t_i} + 2\phi_C(1 - \phi_C)JC_{t_i} + (\phi_X\phi_C(1 - \phi_X)(1 - \phi_C)JXC_{t_i}) + AXC_{t_i} \quad (5)$$

The sum of probabilities defining all of song crystallization outcomes for the songs of fathers with song type X is:

$$(\phi_X)^2 + (1 - \phi_X)^2 + 2\phi_X(1 - \phi_X) = 1 \quad (6)$$

*Learning curves.* To define how young males' song learning is influenced by the songs they hear, we used learning curves based on type III Holling response curves[59] which provide a means to numerically capture functional responses. In our model, the type III curve models the response of juvenile to the song form of adults in the population based on two variables: (1) frequency-dependent bias that favors one form based on its prevalence within the adult population, and (2) selection that favors a particular form of the song.

The learning curves, $\phi_x$ for the high note cluster and $\phi_c$ for the click train, are modified forms of the type III Holling response curve):

$$\phi_x = \frac{x^\beta/\sigma}{(1 - x)^\beta + (x^\beta/\sigma)} \quad (7)$$

and

$$\phi_c = \frac{\sigma c^\beta}{(1 - c)^\beta + \sigma c^\beta} \quad (8)$$

where x is the proportion of the high note cluster within the population, c is the proportion of the click train within the population, β is frequency-dependent bias (favoring learning the novel or retaining the common variant), and σ is selection on the novel variant (a preference for learning the variant that is not dependent on

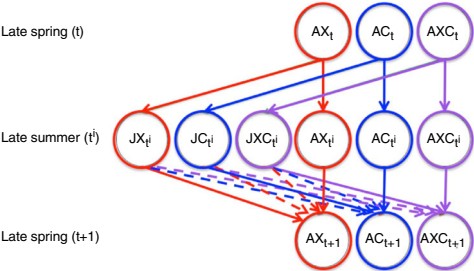

**Fig. 5 Model of song development.** We used two age classes (J = juvenile and A = adult) and three classes of introductions (C = click trains, X = high note clusters, and XC = both). In the late spring of a given year (time = t), only adult males are present. In late summer, those adults have bred and both they and juvenile males are present; at this intermediate time (tⁱ) each male is initially allocated the same introduction type as his father (solid lines). Then, as song development progresses and juvenile males can be influenced by other tutors, they may retain their initial introduction type or switch to either of the other two types (dashed lines) before they crystallize their songs late in the following spring (time = t+1), and join the breeding cohort, which also includes adult males from the previous year who returned to breed again.

frequency of the variant and includes factors such as prestige bias, success bias, status, and content bias). Note that the two learning curves do not have identical equations, because selection is not frequency-dependent. In these equations, $\beta > 1$ corresponds to conformist selection, and when $\beta < 1$ the rare form is favored. Values of $\sigma > 1$ correspond to selection for a novel variant and values of $\sigma < 1$ correspond to selection against a novel variant. The parameters $\beta$ and $\sigma$ allow us to test the relative roles of frequency-dependent bias and cultural selection, as well as various combinations of the two by using a single function giving the probability that social learning will result in a juvenile male crystallizing a particular song variant.

Males that sang both high note clusters and click trains (the AXC class) could be interpreted in one of two ways within this framework:

Two-trait: by counting each variant individually, so that a bird singing both variants is counted twice in calculations of variant frequencies (once for high note clusters, and once for click trains), while a bird singing one form is counted only once. In this scenario, frequencies were calculated as (time subscripts omitted for clarity):

$$P_C = \frac{AC + AXC}{AC + AX + 2AXC} \tag{9}$$

and

$$P_X = \frac{AX + AXC}{AC + AX + 2AXC} \tag{10}$$

Blended trait: each bird was counted once (birds that sang a single variant were weighted twice as much as those that sang both traits). In this scenario, frequencies were calculated as:

$$P_C = \frac{2AC + AXC}{2(AC + AX + AXC)} \tag{11}$$

and

$$P_X = \frac{2AX + AXC}{2(AC + AX + AXC)} \tag{12}$$

*Innovations.* As most males singing click trains in the 1980s and early 1990s also sang a high note cluster, we assumed that the innovators' songs included both forms. We know that click trains first appeared in the population between 1983 and 1987, as they were absent in 1982 recordings and present in 1988 recordings. Prior to 1983, all adults sang high note clusters and so belonged to the AX class. We modeled the appearance of click trains in the population with the term **in**, which represented the number of innovators (which we modeled as entering the population in class AXC, see the next section), and was added in any year from 1983 to 1987. To maintain populations at consistent levels, we subtracted the number of innovators from the AX class in the year the innovation was introduced.

*Choice of values for innovators and years.* First, we assumed that interstitial notes, whether high note clusters, click trains, or both, represented a single trait. We tested this assumption by running the model with either (1) the blended trait or (2) treating click trains and high note clusters as two distinct traits (see Supplementary Table 2 and Supplementary Fig. 2); the blended trait model fit the data better.

We know from the corpus of recordings that click trains were not observed in 1980 or 1982, when high note clusters were the prevalent form. Click trains were first recorded in 1988. Because we do not have recordings for the period spanning 1983 to 1987, each of these years is potentially the time of the initial introduction. We used the earliest possible year, 1983, as the default, because we observed potential precursors of the click train in 1982 songs. We also modeled the appearance of initial innovations for the years 1984 through 1987 (Supplementary Table 3 and Supplementary Fig. 3).

The number of innovators (individuals that sang the click train in the first year it appeared on the study site) is unknown. We chose a default value of 2 males (2.9% of the study population of 70) for two reasons. First, innovations we have observed in other segments of Savannah sparrow songs initially appeared in the songs of 2 or 3 individuals. Second, this "mutation rate", $\mu = 0.029$ per song per year, is in the range found in previous work on the introduction of innovations in learned songs: 0.001 to 0.035 per year in U.K. chaffinches[85], and ~ 0.057 in New Zealand chaffinches[86] This value is also in the middle of the range used to model human cultural evolution (0.004 to 0.128)[87]. We varied the number of innovators from 1 to 8 ($\mu = 0.014$ to $\mu = 0114$) to assess the effect of this parameter on the model's results (see Supplementary Table 4 and Supplementary Fig. 4).

Our models thus used, as default values, two innovators, appearing in 1983, that sang both click trains and high note clusters as a blended trait, and we tested the effects on the modeling results by varying these default values.

*Implementation and evaluation.* The model was implemented in the R[88] package POMP[89] (Partially Observed Markov Processes), using embedded C code. We performed a grid search over a range of the parameters $\sigma$ and $\beta$ (from 0.5 to 2.0 in 0.05 steps for each parameter if not otherwise stated) and calculated the estimated log likelihood for each parameter combination. We used an initial

burn-in of 50 years prior to the first year for which we compared the model to existing data (1980). We repeated this analysis for each set of initial conditions (year the innovation was introduced, and blended vs. two-trait categorization for birds that sang both high note clusters and click trains). We visualized the model space with heat map plots prepared using MatLab, and identified the maximum likelihood estimate (MLE) and the corresponding 95% confidence intervals. Using the best fit parameters (those that corresponded to the MLE), we then ran the model again 50 times to generate average and 95% CI trajectories for frequencies of song variants and plotted them in the same manner as the observed field data.

**Song playback study**. We tested the responses of Savannah sparrows on their territories in early July of 2011 (when most pairs were feeding young or beginning a second clutch) to song segments with click trains that included different numbers of clicks. None of the songs of 39 birds recorded on the study site in 2011 included high note clusters. The mean number of clicks within click trains was 3.93, ranging from 0 (3 birds) to 7 (3 birds), with a mode of 4 clicks in a train ($n = 16$). All of the subjects of the playback study would have had the opportunity to hear click trains ranging from 0 to 7 clicks, but would not have been familiar with high note clusters. Because comparisons of responses to songs with click trains and high note clusters would have been confounded by the issue of familiarity, we only tested subjects' responses to the number of clicks in a train. (A test of the efficacy of click trains and high note clusters in hand-reared birds that had not been exposed to either form might address the question of how preferences may be shaped by social learning).

The stimuli were constructed from high-quality recordings of introductory sections from the songs of 12 different males to produce different 12 stimulus sets, to avoid pseudoreplication. The introductory sections of the twelve songs were originally composed of 5–8 introductory notes, between which 1–3 click trains that included 3–7 clicks. Each of these introductory segments was extracted and then digitally altered (using Audacity, audacityteam.org) to produce a set of four different stimuli that included 0, 2, 4, or 7 clicks in each click train. The introductory notes, the temporal spacing of the introductory notes and the length of the entire introductory segment was the same for each stimulus within a set. Clicks were added to a train by duplicating existing clicks and adjusting them to be evenly spaced within the interval between introductory notes. Clicks were removed by replacing clicks at the end of a train with silence. Since introductory notes are substantially longer (mean = 67 ms) than clicks (mean = 2 ms), a change of one click in a click train stimulus represented a change of, on average, 0.91% in the signal duration (taking into account that adding one click to a train meant adding one click to all instances of that train within a stimulus). Introductory notes are also substantially louder than clicks, and so the overall change in the sound intensity within different stimuli was very small. To the human ear, longer click trains make the intervals between the louder, longer introductory notes sound somewhat "raspier" than shorter click trains, but the difference is subtle.

Each of 25 male subjects was tested with all four stimuli from one set. Each trial started with a "primer", a stimulus consisting of introductory notes without interstitial notes[55]. Two minutes after the bird's response ended, the first test stimulus was presented for two minutes (at 12 second intervals). The next stimuli were presented in succession, with a delay of two minutes after the bird's response ended for each stimulus. Stimuli were presented in a randomized order, and each stimulus set was used at least twice. The response duration and behaviours of males (crouching with head feathers flattened close to the skull, aggressive displays[48] and vocalizations[90]) were noted. We used duration, measured as time from the end of the stimulus presentation until the male ceased responding (defined as moving 20 m or more away from the speaker, or singing a full and loud song, or engaging in feeding or preening behaviour), as our primary measure of male response[55]. Because the strength of the response varied across birds, we normalized response durations for each individual bird in Fig. 4c. To correct for a rightward skew in the distribution, we log-transformed the raw response duration measure and assessed the relationship between response duration and number of clicks ($F_{1,73} = 10.97$, $P < 0.005$), using a generalized mixed-effects model implemented with the lme4 package[91] in R which included the identity of the subject ($F_{24,73} = 3.84$, $P < 0.000001$) as well as the trial order ($F_{1,73} = 0.012$, $P > 0.9$) as random effects. We did not record songs produced during stimulus playback; we observed an average of 0.6 songs per trial, which would not have provided a large enough sample size for analysis.

Females did not always respond to the playback stimuli. When they did respond (in 11 of 25 trials) their responses differed from those of males: females typically stood erect rather than crouching, elevated their crest feathers instead of flattening them, and were never observed to give aggressive wing flutters or vocalizations but rather hopped towards the speaker while peering about alertly. Because female responses to other song stimuli presented in previous studies used the postures and behaviours typical of male aggressive responses, we interpret the approach with an erect posture and crest as having a different valence: investigative/approach rather than aggressive. We noted both which stimuli the females approached and which stimulus they first approached and evaluated the effects of click number with a Chi-squared test.

**Reporting summary**. Further information on research design is available in the Nature Research Reporting Summary linked to this article.

## Data availability

The data generated in this study are provided in the Source Data file. Song recordings are available from the Dryad database at https://doi.org/10.5061/dryad.k98sf7m7x[92]. Source data are provided with this paper.

## Code availability

R scripts for the model and a file with historical song data file are available at https://zenodo.org/record/6643190#.YrITpHbMLIU[93].

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

## Acknowledgements

A grant from the Groff Foundation to Williams College supported the work of H.W., A.S., and A.R.R. Grants from the Natural Sciences and Engineering Research Council of Canada (NSERC) supported the work of D.J.M, D.R.N, A.E.M.N, and S.M.D. Ron Bassar commented on an early version of this paper. We thank Clara Dixon for song recordings from 1980 and 1982; Nat Wheelwright for the demographic data he collected during the period from 1988–2004 and song recordings made prior to 2003; and Iris Levin for 2003 song recordings. We thank Bowdoin College for logistical support; this is contribution #289 from the Bowdoin Scientific Station.

## Author contributions

H.W. and D.J.M. recorded songs. H.W. analysed songs and performed playback experiments. J.C.B., A.R.R., and A.S. developed and coded the model, and J.C.B. and H.W. edited the code. D.R.N., A.E.M.N., D.J.M and S.M.D. collected demographic data. H.W. wrote the paper and all authors contributed to editing and rewriting the paper.

## Competing interests

The authors declare no competing interests.
