## [Peer Review File · Nature Communications]

Cumulative Cultural Evolution and Mechanisms for Cultural Selection in Wild Bird SongsReviewers' Comments:

Reviewer #1:

Remarks to the Author:

This paper provides interesting evidence of cumulative cultural evolution in the songs of savannah sparrows. Given the paucity of robust evidence for CCE in wild animals, this is an important contribution to the literature. Specifically, the authors argue that over a period of 35 years one vocal feature was replaced with another, which (1) elicits stronger responses in listeners and (2) is associated with elevated reproductive success. The authors also suggest that the change in songs results from "cultural selection" rather than "frequency-based learning bias".

I found the evidence for CCE fairly convincing (but see below), but I must admit I found the manuscript rather convoluted and hard to follow. In particular, the opening sections sometimes seem to conflate cultural evolution in general with cumulative cultural evolution specifically. This makes it difficult for readers to discern precisely what question the paper is seeking to address, and how it adds to the existing literature. These issues could be addressed fairly easily by defining CCE at the start and then carefully setting out how the results fit the criteria outlined in reference 3. Providing a definition of CCE early on will also help to clarify the difference between CCE and well-established examples of cultural evolution in bird and cetacean song.

Below I detail my two main queries and a number of more specific issues:

1) How convincing (and novel) is the evidence for CCE?

There is clearly cultural evolution at play here, but is it cumulative cultural evolution (sensu ref 3)? The argument here rests on evidence that (a) one socially learned aspect of song was replaced with another (following an S shaped trajectory characteristic of cultural transmission) and (b) this change is associated with an increase in reproductive fitness. (a) Seems clear and convincing but my understanding is that the pattern of change in song features has been described in previous work (e.g. ref 47). For (b) the central piece of evidence is that when both song features were prevalent in the population males that sang click trains fledged more young than did males that sang high note clusters. However, this also seems to have been reported in previous work [47, 49]. This led me to wonder (perhaps unfairly – please do correct me if I'm wrong) whether the paper rests more on re-packaging of old results than on new insights. Clarifying precisely what previous work has shown and whether and how new results add to this could help.

One novel element in the current paper is the playback experiment suggesting that both male and female listeners responded more strongly to longer click trains. To understand this result, I would like to know whether the birds that participated in the experiment had previously heard click trains of varying lengths. If, for example the majority of the population now sings 7-click trains then the weaker responses to shorter trains could simply be because audiences have not heard them before. Are there any data to allow you to distinguish between (i) stronger responses to longer click trains vs (ii) stronger responses to more familiar/more common call features?

2) Are changes driven by "Cultural selection"?

The authors compared mathematical models incorporating either "frequency dependent learning biases" or "frequency-independent biases that favor a particular form, which we call selection on that form." It is not at all clear to me why frequency-dependent learning biases are not labelled "selection" but all other biases are. This lack of clarity makes it difficult to interpret the argument that changes are driven by "cultural selection".

I also have some other questions about the modelling approach. Was prior knowledge of patterns of song learning in this population (e.g. according to ref 49 most males do not learn mainly from their social father) incorporated into the models? If not, why not? Also it seems that while males may boost

their reproductive success by adopting click trains, their (social) sons do not necessarily inherit their song (c.f. ref 49). What does this dissociation between the apparent adaptive benefit of the trait and its heritability mean for the potential for CCE?

Specific comments:

L12: The definition of cultural evolution in the abstract is rather vague

L18: The evidence for CCE in non-human animals seems exaggerated here. Arguably only Sasaki & Biro's study on pigeons (and perhaps Feher et al's work on zebra finches) provide solid experimental evidence.

L18: This sentence also cites work on wild cetaceans, yet in the next sentence you say there is no evidence from wild animal populations.

L105: The model assumes two innovators, but this is at best an informed guess. What happens if you vary the number of innovators in the model?

Main text: There often seems to be a mismatch between the statistics reported in the main text and the extended data tables. For instance, on line 124 you state: "The version of this model with the lowest AIC value had a moderate rare form bias (0.76) and only a poor fit to the historical data (DAIC = 67.6; Extended Data, Table 1)." However, according to Table 1, the lowest AIC is for the "Selection only" and "Selection and bias" models, with AIC = 45.2. The estimate given for the "Bias only" model is 0.74, not 0.76.

Similarly L 138 gives an AIC value of 20.6 and refers to Table 1, but none of the models in this table show this AIC value.

L 148: "This result points out the importance of cultural selection – and the absence of a role for frequency-dependent learning bias – in the replacement of high note clusters by click trains in the Savannah sparrows' songs." This statement seems at odds with the fact that the "Selection and bias" model has an identical AIC value to the "Selection only" model in Table 1.

L188-197: Why are there no formal statistical analyses of the evidence presented here?

Figure 4C: Rather than just showing means and SEs, it would be useful to also show the raw datapoints here. This is important to give readers an impression of the variation in the data and the strength of the effects.

Methods for song playback study: More detail is needed here. In particular, please provide information on the number of discrete calls and different callers that contributed to each playback track so as to reassure readers that playbacks do not suffer from pseudoreplication. If songs from the same caller appear in different playback sets, how did you control for this? Please also state the data distribution used for the mixed model. It would also be useful to provide information on the focal individuals' prior experience of different lengths of click trains or, at least on how common each length of click train was in the population at the time of the experiment (see earlier comment).

Extended data: It would be clearer if models in tables were ordered by AIC. Tables should also include errors and confidence intervals associated with each effect size estimate.

Reviewer #2:

Remarks to the Author:

This article compiles an impressive 35 year dataset of territorial song in savannah sparrows to describe the change of part of the song over this period to a novel type, followed by change in this

song type itself. The authors attempt to uncover the likely process and mechanisms driving this change by building a time dynamical model where they vary parameters for innovation, strength of selection and learning biases. Finally, they perform a playback experiment to try and ascertain whether individuals respond differently to song with fewer or more “clicks” (components of the novel song type), and use this to infer why it might have been selected for (male competition or female choice).

There is certainly a lot of work and a diversity of components packed into this manuscript, as I stated above, it is an impressive undertaking, and I commend the authors. I want to start by clearly stating that I am not a modelling expert; so while the model appears to be reasonable to me, I can't review it in depth. I have therefore focused my review on the other parts of the manuscript.

1) My major concern with this work is in its theoretical underpinnings.

It attempts to sit between four related fields: bird song, language evolution, cumulative cultural evolution (CCE) and animal culture. While this is an interesting endeavour, it is also difficult, and in my view the manuscript never really makes a strong convincing argument for any of them. This is most evident in the introduction: in the first paragraph there appears to be some confusion between culture and cultural evolution, and CCE, none of which are clearly defined. Sometimes in later paragraphs they are even used interchangeably. There is little time spent on the rich literature from cultural evolution in song. Rather than the somewhat tangential discussion of language evolution in humans, the authors could be more explicitly discussing previous work showing cultural evolution in bird and whale song. This includes Lachlan et al (2018) (arguing that conformity is the reason for strong song stability over time in sparrow song), and Garland et al. (2011) and Otter et al 2020 showing the rapid spread and replacement of song elements in populations of birds and whales. I should mention that Otter et al. (2020) was probably not published at the time of writing this manuscript, but discussion of their results should certainly be included in a revised version.

More importantly, on the argument for CCE in this manuscript: I think the authors need to reassess their evidence more carefully with respect to the existing literature and theory. In my opinion they have evidence for cultural evolution, perhaps even for the potential precursors of CCE, but no clear argument for full CCE, at least from the data they propose shows it. CCE depends on either 1) recombination of existing traits and/or 2) the modification of traits through the incorporation of innovations to increase complexity (and perhaps efficiency) over generations. First, a novel song type is observed in the population, and is shared with the existing song type. Their model suggests a blended version of the song is being used before it dominates and the other type is mostly lost. Their claim for CCE then hangs on the final part of their dataset, showing that once the novel song type “click train” has dominated, it increases in click number (and in variability in number of clicks). Using this definition, the best evidence here for CCE actually comes from the blending of the two song types after introduction (recombination). The subsequent loss of this pre-established song type reveals a fascinating gain and loss of complexity that is currently underexplored in the manuscript, perhaps suggesting that there is a limit to the level of complexity in song that can be maintained by this population.

The authors suggest that the increasing number of clicks is evidence of CCE. But they also show that the variation in click number is increasing over time, and suggest that males may be modifying their song to sound more novel by improvising (hence leading to increasing number of clicks overall in the population). However, without showing that the elaborated form is socially learned, there is no evidence for cultural selection and CCE. We need to know whether the general population is moving towards higher number of clicks over generations (without lower click sequences also being maintained) – evidence for this might come, for example, from a tendency for first-year males to sing songs with more clicks than older males. Given your other analysis on the song of young males (that they were more likely to sing clicks than the other song type), it seems that this analysis would be possible. An opposite trend (old males singing with more clicks) would suggest individual modification,

and no CCE.

2) My second major concern is with the playback experiment. In particular, I have two questions about the methodology. First – the methods section suggest that the clicks were replaced with silence. Therefore, unless I've misinterpreted, playbacks with more clicks had more overall sound. Was there a control to ensure that the birds weren't responding more to this general heightened stimulus, e.g. by also having a playback condition where another part of the song was repeated? Second – it seems to me that this playback was a missed opportunity to play and compare the three song types – high note clusters, the blended song, and pure click trains. A differential response to them would have confirmed and extended the earlier results in line 202-212 (ie. whether click trains were adopted by males because they were favoured by females or because they are easier to learn or retain).

For the rest of the manuscript, I have line-by-line comments, some of which overlap with the above:

L 31 – this sentence seems to confuse culture with cultural evolution. Your examples also are a mix of the two (geographic variation in chimpanzees, vs. population-shift in whale vocalisations). I would suggest keeping these clear and starting with culture, then more clearly defining cultural evolution at L45.

L50 – The introduction moves from cultural evolution to the specific example of language evolution. Wouldn't it be clearer to move to the example of vocal cultures in animals, specifically song in birds and whales? There is a lack of discussion of cultural evolution in bird song in this introduction (a rich field, as I'm sure you know well).

L70-72 – While the example from Hastings is poetic, surely there are better examples from bird and whale song? If you want to keep it, then you should at least also compare and contrast with previous examples from bird song.

L105-108 – The choice of parameter here needs much more justification. Why wouldn't you just take the parameters from the two previous innovations in birds? I don't see how could generalize learning error rates from vastly different behaviour domains like naming conventions and pottery.

L127 – how large was the population? Did you consider whether some of the effects could result from stochasticity, e.g. random loss of most of the high rote singers, then internal or cultural selection for simplicity?

L161 – Is this value representing a weak positive or weak negative frequency dependent bias?

L182 – what about song sharing on the wintering grounds, as suggested by Otter et al (2020)?

L224 – how is this variation distributed, are males consistent, or does the one individual sing songs with variable numbers of clicks?

L250 – this is not exactly how I would define cultural selection. Rather I would say it is driven by learning biases and internal reinforcement.

L256 – Personally, I wouldn't say that guided variation generates variation, rather it narrows variation, by only allowing cultural selection to operate in particular directions.

L261 – This is where your results depart from expectation (increasing variation). Not sure this argues in favour of directed cultural selection though, rather than working alongside it as another process. I think you need to develop this argument more.

L282 – why do you specifically point to oblique and horizontal transmission? It seems funny to raise

this in the last sentence, when the relative merits or importance of both transmission modes happening in tandem for CCE haven't been discussed (unless I missed it) elsewhere in the manuscript.

Figure captions 1 & 2 – these are very long, I would suggest revising to be more concise.

L504 – Do you have any information on the immigrants, do they differ in their song?

L633 – can you explain more about these potential precursors? This seems potentially very interesting! (Maybe I missed it in the main text, if so apologies).

L666 – what was the sung response? Did it differ in any way from the song that those males usually sung?

Reviewer #3:

Remarks to the Author:

Please see associated pdf file.

Review of "Directional Selection and Cumulative Cultural Evolution in a Wild Songbird"

This study contributes to an expanding body of work on how social learning biases influence the cultural evolution of populations, and provides an alternative method of generative inference using discrete time modeling for simpler cultural systems (i.e. one variant replacing another, rather than a large number of variants [Lachlan et al., 2018, *Nature Comms*]). The model appears to be well-executed and incorporates important life-history information about the species. Additionally, this study is the first to follow up this kind of simulation-based work with playback experiments, to demonstrate that the new variant spreading through the population elicits greater responses from both females and males, and with assessments of the reproductive success of males singing particular variants. Overall, I think it represents an important advance in cultural evolution in non-human animals.

That being said, I have several critiques, two of which are major conceptual points.

First, the study is framed as the first "direct evidence of naturally occurring cumulative cultural evolution in a wild animal population". The results of this study meet the criteria for cumulative culture evolution (CCE) proposed by Mesoudi and Thornton (2018, *Proc Roy Soc B*), but according to those criteria, previous studies have already provided evidence for CCE, for instance in pigeons (Sasaki & Biro, 2017, *Nature Comms*) and zebra finches (Feher et al., 2009, *Nature*). In addition, these criteria appear to be too conservative, as they basically describe exactly what we would expect from cultural evolving traits that are sexually selected. This study demonstrates that social learning biases (in this case content-based bias, which the authors refer to as "cultural selection" [see below]) can drive the replacement of a less attractive cultural variant with a more attractive one, which does not seem to provide any new evidence for CCE. The majority of papers on the topic use a more strict definition for CCE that requires an increase in either complexity or efficiency (Dean et al., 2014, *Biol Revs*; Schofield et al., 2018, *Primates*), neither of which are demonstrated here. I feel that framing the study around cumulative cultural evolution is inaccurate and distracts from the core finding about content-based bias. My recommendation would be to remove all references to cumulative cultural evolution entirely.

Second, the authors do not present a clear notion of "cultural selection", which according to lines 92-94 appears to be content-based bias, or a preference for traits based on some aspect of their content. Throughout the paper, the authors make a dichotomy between "learning bias" and "selection", even though, in their own words, frequency-based bias is also a selective process (see lines 46-48). The more accurate framing would be frequency-based bias vs. content-based bias, as used by Lachlan et al. (2018, *Nature Comms*). Rendell et al. (2011, *Trends Cogn Sci*) have a great diagram clarifying the different forms of social learning biases.

Other points of criticism are:

- The first sentence of the abstract has several problems. Cultural evolution is the change in socially learned behavior, not the learning (transmission) of behavior itself. Only some learned

behavior is socially learned and therefore relevant to culture. Unfortunately the authors' concept of culture does not specify social learning. This leads to further problems interpreting the rest of the sentence: what does it mean for learning to happen within and across lineages and generations? The subtext one must gather is that the learner is one individual and the one learned from is another individual, and these can be in the same or different lineages and generations, but none of this is clear from the sentence.

- The last sentence of the abstract has the word driver twice. It appears that the authors mean that sexual selection is the major driver. Cumulative cultural evolution is not a driver of change, it is a description of the change itself.

-Why did innovation rate assume a single innovation rate, rather than including it as a parameter with a distribution informed by the cited studies? Also, varying the mutation rate and numbers of innovators would permit the researchers to see how much their assumptions shape the results.

-The authors cite a previous paper (2013) from the same research group showing click trains replacing high note clusters. They actually cite this paper as the source of this result, despite claiming in their abstract that it was a major result of the present paper. This is not a minor point, because a main claim, even in the title of the paper, depends on this being novel. The present paper cannot claim that cumulative cultural evolution has never been documented if it was documented in this very population in 2013 (at the very least—in fact it has been documented elsewhere as well, and their mention of “cumulative” adds nothing to the interpretation of their results or to the paper’s novelty, as I mentioned above. The interesting thing about this study is really something else: its identification of the mechanism driving the change.)

-The ending (last sentence) is hasty, like the beginning. There is no such thing as “horizontally across generations”.

-One point that is highlighted by their research but not discussed is the important question of whether the meme is the thing with fitness or the individuals with the “meme”. Their fledging and sexual selection data suggest the latter. Cultural evolutionists have typically presumed the former. I wonder what the authors think.

-The authors' claim that high note clusters are replaced by click trains is a seemingly concrete result that conceals some ambiguity. This issue is demonstrated in Figure 1. In song *a*, a high note cluster is essentially a click train followed by a high note and then a more rapid click train. Song *b* shows that the second click train can vary continuously into what could be called a trill. Regardless, saying that high note clusters were replaced by click trains might be a little too neat for what really happened, since high note clusters appear to include at least one click train. This terminology was set up in their 2013 paper, which has already passed review, but I believe it is important to get the wording and descriptions of their acoustic features clear and (if necessary) correct for this paper, because the distinction between these two forms is foundational for all

of the results in the current paper. Here are some closeups of the spectrograms to make this issue clearer:

From figure 1, song a:

This is the “variable note” section of the high note cluster. It’s followed by a high note, and then by:

This is a “trill” that ends the high note cluster.

This is a “click train”, the completely different trait that replaced the sequence partially described above.

Here three different words (variable note, trill, click train) are being used to describe an acoustic signal that can be in fact very similar or even identical. The click train is said to have replaced the high note cluster in the population, but many high note clusters are comprised partly of one or more click trains. This was not clear in the manuscript, and actually influences the results. The initial “variable note” section could be just a click train, among other things. The “trill” section apparently varies continuously between a discrete click train and a more rapid sequence approaching a buzz. The only categorically distinct aspect of the high note cluster is the intervening “high note”. The authors should deal with this ambiguity either by having a more quantitative way of distinguishing between what they consider to be two different acoustic forms, or else changing the wording and categorizing to reflect the complexity of what really happened to the song over the decades, or both.

Summary of major changes to the manuscript (pages 1-2)

Point-by-point responses to the reviewers' comments are provided on pages 3-17;

- responses to reviewer 1 begin on page 3
- responses to Reviewer 2 begin on page 11
- responses to Reviewer 3 begin on page 17.

We thank the reviewers for their thoughtful and comprehensive comments. Responding to them has greatly improved this manuscript. We have provided detailed explanations of changes made in light of the comments in separate document, but here we summarize the most important changes.

All three reviewers raised questions about the definition of cumulative cultural evolution (CCE) and how and whether our data demonstrate that CCE occurred. Following the suggestion of Reviewer #1, we now use and refer to the definition from Mesoudi and Thornton (2018), which calls for, among the core criteria for cumulative cultural evolution, a) successive changes in a population's behaviour and b) demonstration that the successive changes increase the "efficiency" of the behavior in accomplishing its purpose. We have reframed all the parts of the paper to use this definition, and have organized the results to show how each of the two critical criteria for cumulative cultural evolution (as opposed to cultural evolution) have been met. To satisfy the first criterion, that changes must be successive to be cumulative, we emphasize the analysis of the timing of the increase in the number of clicks in click trains. Our data show that this trend began in 2004, more than 15 years after click trains were first recorded – only after more than 75% of the birds on the study site had adopted click trains. We also show that the increase in click number continued even after high note clusters had disappeared in 2011. To satisfy the second criterion we have the results of the playback experiment, showing that longer click trains were more efficient at eliciting stronger aggressive responses in males and stronger approach responses in females. The revised manuscript makes these points explicit and links them to the criteria for cumulative cultural evolution.

Reviewers 1 and 3 found the terminology and framing of types of evolutionary mechanisms to be problematic. Following the suggestion of Reviewer 3, we have used Rendell et al.'s (2011) scheme for defining three types of mechanisms that result in cultural evolution: 1) drift (random or stochastic processes), 2) frequency-dependent selection (copying depends only on the prevalence of a learned form of a behavior within the population, and 3) direct selection (which includes, among other behavior-related characteristics, demonstrator bias, payoff bias, and sensory predispositions). We used this terminology consistently throughout the paper, which also makes the discussion of mechanisms much clearer.

Reviewers 1 and 2 asked for additional information about the playback experiment. We have added information about a) the sound levels and amount of sound in different stimuli (adding a click adds only very little to the length or intensity of the sound in the stimulus), b) the subjects' experience of click train lengths (which corresponded to the stimuli used), c) that we used multiple stimulus sets to avoid pseudoreplication, and d) the reason for not performing playbacks comparing responses to click trains and high note clusters: in 2011 (and in subsequent years), when we performed the playback experiments, no birds on the study site sang high note clusters and such a study would have been confounded by the fact that the two stimulus types would have been familiar / unfamiliar.

Reviewer 3 was concerned about the ambiguity of note types and assignments of songs to click trains and high note clusters. We have added a new figure (Extended Data, Figure 1) showing the basis for categorizing note types, and have added new material to address this point.

Reviewer 3 also asked what was novel about this study, given that the replacement of high note clusters by click trains and the correlation between reproductive success (number of fledglings) and singing click trains had already been published in our 2013 paper. We have now more carefully delineated what was old and what is new in this manuscript. The novel findings are:

- 1) The modeling approach demonstrates that direct selection – and not drift or frequency-dependent selection – is responsible for click trains replacing high note clusters. Although previous studies have documented the role of frequency based selection in maintaining conformity (in swamp sparrows, Lachlan et al. 2018) and drift in songs that change frequently (chestnut-sided warblers, Byers et al., 2010), examples of direct selection in wild animal cultural evolution are rare (although they have been proposed to exist in the abstract, as in Mesoudi, 2021).
- 2) The existence of cumulative cultural evolution (CCE) in a wild animal population. In this manuscript we show that a) click trains evolved in two distinct steps, first replacing high note clusters and later elaborating as more clicks were added, and b) that click trains with more clicks are more “efficient” in eliciting responses from conspecifics, via our playback experiment. Thus we show that this wild bird population spontaneously satisfied the criteria for CCE (see Mesoudi and Thornton, 2018).

All of the changes to the text are in blue type within the revised ms., and all of the substantive changes are described in detail in our response to the reviewers.

Reviewer #1 (Remarks to the Author):

This paper provides interesting evidence of cumulative cultural evolution in the songs of savannah sparrows. Given the paucity of robust evidence for CCE in wild animals, this is an important contribution to the literature. Specifically, the authors argue that over a period of 35 years one vocal feature was replaced with another, which (1) elicits stronger responses in listeners and (2) is associated with elevated reproductive success. The authors also suggest that the change in songs results from “cultural selection” rather than “frequency-based learning bias”.

I found the evidence for CCE fairly convincing (but see below), but I must admit I found the manuscript rather convoluted and hard to follow. In particular, the opening sections sometimes seem to conflate cultural evolution in general with cumulative cultural evolution specifically. This makes it difficult for readers to discern precisely what question the paper is seeking to address, and how it adds to the existing literature. These issues could be addressed fairly easily by defining CCE at the start and then carefully setting out how the results fit the criteria outlined in reference 3. Providing a definition of CCE early on will also help to clarify the difference between CCE and well-established examples of cultural evolution in bird and cetacean song.

We have reframed the introduction to provide a clearer account of the definition of and criteria for cumulative cultural evolution, including the points that a) there must be a time difference in the two sets of changes and b) the second change must increase the effectiveness of the signal, and we have made the links between our data and these criteria explicit in the discussion.

Below I detail my two main queries and a number of more specific issues:

1) How convincing (and novel) is the evidence for CCE?

There is clearly cultural evolution at play here, but is it cumulative cultural evolution (sensu ref 3)? The argument here rests on evidence that (a) one socially learned aspect of song was replaced with another (following an S shaped trajectory characteristic of cultural transmission) and (b) this change is associated with an increase in reproductive fitness. (a) Seems clear and convincing but my understanding is that the pattern of change in song features has been described in previous work (e.g. ref 47). For (b) the central piece of evidence is that when both song features were prevalent in the population males that sang click trains fledged more young than did males that sang high note clusters. However, this also seems to have been reported in previous work [47, 49]. This led me to wonder (perhaps unfairly – please do correct me if I’m wrong) whether the paper rests more on re-packaging of old results than on new insights. Clarifying precisely what previous work has shown and whether and how new results add to this could help.

The reviewer’s comment points out the importance of clearly framing this paper’s rationale and the important advances it includes, which are: 1) the investigation of the mode of cultural selection via modeling, and 2) the demonstration that the second step of the change a) follows incorporation by a super-majority of individuals, and b) is an elaboration that provides increased efficiency in the signal (playback experiment). These points needed to be better highlighted in the paper, and we thank all three reviewers for

pointing this out. We have made changes to both the introduction and the discussion to address these comments.

2) Are changes driven by “Cultural selection”?

The authors compared mathematical models incorporating either “frequency dependent learning biases” or “frequency-independent biases that favor a particular form, which we call selection on that form.” It is not at all clear to me why frequency-dependent learning biases are not labelled “selection” but all other biases are. This lack of clarity makes it difficult to interpret the argument that changes are driven by “cultural selection”.

The terminology has been changed throughout the paper. The issue arose because there are, essentially, two kinds of learning biases: frequency-based biases, which depend solely upon the distribution of traits within a population, and other biases (content, demonstrator, payoff) which depend upon either the qualities of the song, the qualities of the singer, or the consequences of singing the song. We have used the terminology of Rendell et al., 2011 to distinguish between these two types of selection: a) frequency dependent and b) direct, which are both distinct from drift.

I also have some other questions about the modelling approach. Was prior knowledge of patterns of song learning in this population (e.g. according to ref 49 most males do not learn mainly from their social father) incorporated into the models? If not, why not?

Yes, we explicitly took into account that males may learn songs from models other than their fathers, and may do so during the spring of their first breeding year. We did so by first assigning young birds to the same song type as their fathers, and then re-assigning survivors to one of three song types (click train, high note cluster, or both) according to probabilities defined by the learning parameters in the model (frequency dependence or direct selection). See the Methods, lines 575-581.

Also it seems that while males may boost their reproductive success by adopting click trains, their (social) sons do not necessarily inherit their song (c.f. ref 49). What does this dissociation between the apparent adaptive benefit of the trait and its heritability mean for the potential for CCE?

This point is closely related to the question raised by another reviewer about whether it is the meme or the individual that is under selection. We now address this question in the discussion, pointing out that a male’s “quality”, in terms of survival, is not coupled with the quality of the song, in terms of how likely it is to be copied. We suggest that direct selection on the signal may take the form of either observation of females’ sensory predispositions or demonstrator bias – copying males that are successful in attracting females. (See lines 330-343.)

Methods for song playback study: More detail is needed here. In particular, please provide information on the number of discrete calls and different callers that contributed to each playback track so as to reassure readers that playbacks do not suffer from pseudoreplication. If songs from the same caller appear in different playback sets, how did you control for this?

There were 12 stimulus sets, each derived from the song of one (and only one bird), specifically to avoid pseudoreplication. This information has been clarified in the Playback Methods section (lines 686-690).

Please also state the data distribution used for the mixed model.

We had not corrected for skewness in the response duration measure; this has now been log-transformed (stated in the Playback Methods, lines 712-717) and we now report the results of the model with the log-transformed data in the main body of the paper.

One novel element in the current paper is the playback experiment suggesting that both male and female listeners responded more strongly to longer click trains. To understand this result, I would like to know whether the birds that participated in the experiment had previously heard click trains of varying lengths. If, for example the majority of the population now sings 7-click trains then the weaker responses to shorter trains could simply be because audiences have not heard them before. Are there any data to allow you to distinguish between (i) stronger responses to longer click trains vs (ii) stronger responses to more familiar/more common call features?

During the year when the playback study took place, the 39 recorded songs included click train lengths ranging from 0 ($n = 3$) to 7 ($n = 3$) and averaged 3.93; the mode was 4 clicks per train ($n = 16$). Thus the playback stimulus range corresponded to the click train lengths sung on the study site that year, and so birds had the opportunity to hear the entire range used in the stimuli. (This information has been added to the Playback Methods section, lines 679-683, and, briefly, to the results, lines 253-255.)

If responses to more common features elicited a greater response, the strongest responses should have been to 4-click trains; if less common features elicited stronger responses, the songs with 0- and 7- click trains should have elicited similar responses. (This information has been added to the discussion, lines 293-296.)

Main text: There often seems to be a mismatch between the statistics reported in the main text and the extended data tables. For instance, on line 124 you state: “The version of this model with the lowest AIC value had a moderate rare form bias (0.76) and only a poor fit to the historical data (DAIC = 67.6; Extended Data, Table 1).” However, according to Table 1, the lowest AIC is for the “Selection only” and “Selection and bias” models, with AIC = 45.2. The estimate given for the “Bias only” model is 0.74, not 0.76.

Similarly L 138 gives an AIC value of 20.6 and refers to Table 1, but none of the models in this table show this AIC value.

The values in the Tables were correct values for the parameters b and s (now changed to β and σ), and they have been re-checked against those reported in figures and in the text and corrected. All the AIC values reported in the text have expressed as Δ AIC values and re-checked. We thank the reviewer for catching these errors.

Extended data: It would be clearer if models in tables were ordered by AIC. Tables should also include errors and confidence intervals associated with each effect size estimate.

Table 1 has been re-ordered by AIC. Tables 2 and 3 were already ordered by AIC. Table 4 is ordered by the number of innovators (which is the inverse of AIC order, so the order is systematic).

L12: The definition of cultural evolution in the abstract is rather vague

The abstract has been recast and is now much shorter; definitions are found in the introduction.

L18: The evidence for CCE in non-human animals seems exaggerated here. Arguably only Sasaki & Biro's study on pigeons (and perhaps Feher et al's work on zebra finches) provide solid experimental evidence.

Jesmer et al.'s [19] study of ungulate migrations is conceptually very similar to Sasaki and Biro's work with homing pigeons, and so this reference has been included as well. All of these studies use captive or managed populations, as does Schofield et al.'s re-examination of Japanese macaque feeding which provides suggestive observational evidence for CCE [25].

L18: This sentence also cites work on wild cetaceans, yet in the next sentence you say there is no evidence from wild animal populations.

This sentence has been rewritten and the citations adjusted to correctly state and cite the distinction between experimental studies on captive populations and studies of naturally occurring behaviors in wild populations.

L 148: "This result points out the importance of cultural selection – and the absence of a role for frequency-dependent learning bias – in the replacement of high note clusters by click trains in the Savannah sparrows' songs." This statement seems at odds with the fact that the "Selection and bias" model has an identical AIC value to the "Selection only" model in Table 1.

We have changed the wording lines in 161-164 to point out more clearly that the results of the "full" model are essentially identical to the "direct selection" model precisely because the best version of the "full" model had essentially no frequency-dependent selection; $\beta = 0.99$ is extremely close to $\beta = 1$, which is neutral (neither conformity nor rare advantage).

L105: The model assumes two innovators, but this is at best an informed guess. What happens if you vary the number of innovators in the model?

Larger numbers of innovators appear to be biologically unrealistic both in terms of our own observations and in others' data, so we chose to use two innovators as our default within the model. We also ran the model with a range of innovators, but did not initially include this result in the paper out of concerns about length. The figure and table reporting this information have been put into the Extended Data section (Table 4 and Figure 4), and the outcome summarized in the results section in the same paragraph where we consider year of introduction (lines 178-181). Briefly, a larger number of innovators does result in a better fit to the data. However, increasing the number of innovators had only very minor effects on the values for conformity and selection.

L188-197: Why are there no formal statistical analyses of the evidence presented here?

The results in this paragraph, on survival, have been reformatted and the paragraph has been re-written to include statistical analyses (lines 213-219), and a fifth figure has been added to the Extended Data.

Figure 4C: Rather than just showing means and SEs, it would be useful to also show the raw datapoints here. This is important to give readers an impression of the variation in the data and the strength of the effects.

This has been done for Figure 4a as well as Figure 4c.

Reviewer #2 (Remarks to the Author):

This article compiles an impressive 35-year dataset of territorial song in savannah sparrows to describe the change of part of the song over this period to a novel type, followed by change in this song type itself. The authors attempt to uncover the likely process and mechanisms driving this change by building a time dynamical model where they vary parameters for innovation, strength of selection and learning biases. Finally, they perform a playback experiment to try and ascertain whether individuals respond differently to song with fewer or more “clicks” (components of the novel song type), and use this to infer why it might have been selected for (male competition or female choice).

There is certainly a lot of work and a diversity of components packed into this manuscript, as I stated above, it is an impressive undertaking, and I commend the authors. I want to start by clearly stating that I am not a modelling expert; so while the model appears to be reasonable to me, I can't review it in depth. I have therefore focused my review on the other parts of the manuscript.

1) My major concern with this work is in its theoretical underpinnings.

It attempts to sit between four related fields: bird song, language evolution, cumulative cultural evolution (CCE) and animal culture. While this is an interesting endeavour, it is also difficult, and in my view the manuscript never really makes a strong convincing argument for any of them. This is most evident in the introduction: in the first paragraph there appears to be some confusion between culture and cultural evolution, and CCE, none of which are clearly defined. Sometimes in later paragraphs they are even used interchangeably. There is little time spent on the rich literature from cultural evolution in song. Rather than the somewhat tangential discussion of language evolution in humans, the authors could be more explicitly discussing previous work showing cultural evolution in bird and whale song. This includes Lachlan et al (2018) (arguing that conformity is the reason for strong song stability over time in sparrow song), and Garland et al. (2011) and Otter et al 2020 showing the rapid spread and replacement of song elements in populations of birds and whales. I should mention that Otter et al. (2020) was probably not published at the time of writing this manuscript, but discussion of their results should certainly be included in a revised version.

These are valuable points, and led us to recast the introduction, starting with cultural evolution (and examples from animal systems), followed by a paragraph on mechanisms with examples from bird song (including explicit reference to the results of Lachlan et al.'s 2018 and Otter et al.'s 2020 paper).

More importantly, on the argument for CCE in this manuscript: I think the authors need to reassess their evidence more carefully with respect to the existing literature and theory. In my opinion they have evidence for cultural evolution, perhaps even for the potential precursors of CCE, but no clear argument for full CCE, at least from the data they propose shows it. CCE depends on either 1) recombination of existing traits and/or 2) the modification of traits through the incorporation of innovations to increase complexity (and perhaps efficiency) over generations. First, a novel song type is observed in the population, and is shared with the existing song type. Their model suggests a blended version of the song is being used before it

dominants and the other type is mostly lost. Their claim for CCE then hangs on the final part of their dataset, showing that once the novel song type “click train” has dominated, it increases in click number (and in variability in number of clicks). Using this definition, the best evidence here for CCE actually comes from the blending of the two song types after introduction (recombination). The subsequent loss of this pre-established song type reveals a fascinating gain and loss of complexity that is currently underexplored in the manuscript, perhaps suggesting that there is a limit to the level of complexity in song that can be maintained by this population.

This comment, and related points made by the other reviewers, led us to clarify our presentation of CCE in the introduction (the new third paragraph of that section) and in the discussion. We now explicitly point out that CCE requires showing that a sequence of changes are separated in time, and that elaborated trait is also more efficient.

In the results, we document first that the increase in the number of clicks came only after at least 75% of the birds were singing songs that included click trains (sequential changes) and that the elaboration of adding more clicks to a train made the signal more salient to males and was more efficient at attracting the attention of females, meeting the criteria for CCE as defined by Mesoudi and Thornton.

The authors suggest that the increasing number of clicks is evidence of CCE. But they also show that the variation in click number is increasing over time, and suggest that males may be modifying their song to sound more novel by improvising (hence leading to increasing number of clicks overall in the population). However, without showing that the elaborated form is socially learned, there is no evidence for cultural selection and CCE. We need to know whether the general population is moving towards higher number of clicks over generations (without lower click sequences also being maintained) – evidence for this might come, for example, from a tendency for first-year males to sing songs with more clicks than older males. Given your other analysis on the song of young males (that they were more likely to sing clicks than the other song type), it seems that this analysis would be possible. An opposite trend (old males singing with more clicks) would suggest individual modification, and no CCE.

This question was indirectly addressed in the original ms. (the survivor analysis) and has now been given more prominence, with an additional paragraph in the Results section (lines 239-249) and an additional figure (Extended Data Figure 6). Briefly, first-year males averaged 0.27 more clicks in their click trains; in a model that also included year, this difference was significant ($p < 0.005$). Adult males only very rarely changed the number of clicks in a train.

My second major concern is with the playback experiment. In particular, I have two questions about the methodology. First – the methods section suggest that the clicks were replaced with silence. Therefore, unless I’ve misinterpreted, playbacks with more clicks had more overall sound. Was there a control to ensure that the birds weren’t responding more this general heightened stimulus, e.g. by also having a playback condition where another part of the song was repeated?

Since introductory notes are substantially longer (mean = 67 ms) than clicks (mean = 2 ms), a change of one click in a click train stimulus represented a change of, on average, 0.91% in the signal duration (taking into account that adding one click to a train meant adding one click to all instances of that train within a stimulus). Introductory notes are also substantially louder than clicks, and so the overall change in the amount of the sound

within different stimuli was small. (This information has been added to the Playback Methods section, lines 696-703.)

Second – it seems to me that this playback was a missed opportunity to play and compare the three song types – high note clusters, the blended song, and pure click trains. A differential response to them would have confirmed and extended the earlier results in line 202-212 (ie. whether click trains were adopted by males because they were favoured by females or because they easier to learn or retain).

No birds recorded on the study site in 2011, when the playback experiment was performed, sang high note clusters. Because comparisons of responses to songs with click trains and high note clusters would have been confounded by the issue of familiarity with the song components, we only tested subjects' responses to the number of clicks in a train (this information has been added to the Playback Methods section, lines 696-701).

L 31 – this sentence seems to confuse culture with cultural evolution. Your examples also are a mix of the two (geographic variation in chimpanzees, vs. population-shift in whale vocalisations). I would suggest keeping these clear and starting with culture, then more clearly defining cultural evolution at L45.

Good point. We have tried to make the distinction more explicit, and have noted that geographic variation in a socially learned behavior is indirect evidence for cultural evolution, not cultural evolution itself.

L50 – The introduction moves from cultural evolution to the specific example of language evolution. Wouldn't it be clearer to move to the example of vocal cultures in animals, specifically song in birds and whales? There is a lack of discussion of cultural evolution in bird song in this introduction (a rich field, as I'm sure you know well).

The introduction has been recast in light of this and other comments, and references to more explicit examples from the bird song literature have been included in the second paragraph. The literature on vocal cultures in animals is indeed extensive, enough so that it can and has provided material for several long reviews; in the introduction and discussion we have been forced to touch on only the some of the highlights.

L70-72 – While the example from Hastings is poetic, surely there are better examples from bird and whale song? If you want to keep it, then you should at least also compare and contrast with previous examples from bird song.

We accept the reviewer's suggestion to focus the introduction more tightly on birds and whales, as the best-documented animal models of vocal learning, and the revised manuscript downplays speech/language examples and focuses more upon animal vocal culture.

L105-108 – The choice of parameter here needs much more justification. Why wouldn't you just take the parameters from the two previous innovations in birds? I don't see how could generalize learning error rates from vastly different behaviour domains like naming conventions and pottery.

We have expanded our explanation for the initial choice of number of innovators (see the Methods section, lines 648-657), and have also included an analysis of the model's results with different numbers of innovators (lines 178-182, and Extended Data Table 4 and Figure 4).

L127 – how large was the population? Did you consider whether some of the effects could result from stochasticity, e.g. random loss of most of the high note singers, then internal or cultural selection for simplicity?

The Three Islands population includes approximately 350-500 males. Our core study area and our song recordings followed 50-75 males within that population, or 10-20% (added to methods, lines 499-509).

The “frequency-dependent copying” hypothesis, panel b in Figure 2, is effectively a random copying / loss / cultural drift hypothesis. Because we know that songs are learned, random copying would result in learning of features in proportion to their presence in the population. To include random fluctuations in survival between breeding years that might result in drift, the model includes a stochastic element in separate calculations of the survival rate for adults singing each type of song element, as well as a stochastic element in the survival rate for each class of juveniles (which may or may not change their originally learned song feature class).

Since the binomial is computed separately for each song feature class, the survival rates for each class differ randomly. Figure 2b shows the average of model runs; although each run varied in how well it matched the actual data, the overall probability of reproducing the observed data was extremely low, especially compared to models that included selection (Figs. 2d and 2e).

Language to explicitly address the possibility of random/stochastic mechanisms, or cultural drift, has been added to the introduction (line 50) and the results (lines 130 and 134-137) and figures now explicitly the model without either frequency-dependent or direct selection as the drift model.

L161 – Is this value representing a weak positive or weak negative frequency dependent bias?

Values of β less than 1 represent negative frequency-dependent bias (rare-form bias). The language that elicited this question was moved to the Extended Data, Table 3 and Figure 3, and the direction of the bias is stated.

L182 – what about song sharing on the wintering grounds, as suggested by Otter et al (2020)?

We do not know much about singing behavior on the wintering grounds; Wheelwright and Levin did not observe any. We do know from Woodworth et al. that Kent Island birds scatter over the winter, and so could potentially hear songs from other areas. However, we also know from early season recordings and from Mennill et al. that many birds, especially first-year birds, arrive before crystallizing their songs, and that they crystallize songs heard at the breeding site either during their natal year and in the spring of their first adult year (or both). Since we have not observed click trains in recordings from other sites (our own recordings or those in online libraries), it appears that song learning is primarily restricted to the breeding area in this philopatric population. This information has been included in the discussion on the possible origins of click trains (lines 308-314).

L224 – how is this variation distributed, are males consistent, or does the one individual sing songs with variable numbers of clicks?

Early in the season there is variability (especially in the songs of first-year males), but after crystallization the number of clicks is consistent - although birds may occasionally vary the length of their click trains by one click (information added to lines 223-226).

L250 – this is not exactly how I would define cultural selection. Rather I would say it is driven by learning biases and internal reinforcement.

L256 – Personally, I wouldn't say that guided variation generates variation, rather it narrows variation, by only allowing cultural selection to operate in particular directions.

L261 – This is where your results depart from expectation (increasing variation). Not sure this argues in favour of directed cultural selection though, rather than working alongside it as another process. I think you need to develop this argument more.

All excellent points. The paragraph that elicited these three comments has been recast; the term “cultural selection” is no longer defined there, and guided variation is not mentioned. Instead, the reviewer's suggestion that the argument about variation be developed more has been followed. Lines 363-368 now argue that the combination of innovation and social learning allow individuals to escape the constraint of only being able to learn forms already present in the population, and that increased variation is a signature of this process – which provides an “accelerator” for cultural evolution.

L282 – why do you specifically point to oblique and horizontal transmission? It seems funny to raise this in the last sentence, when the relative merits or importance of both transmission modes happening in tandem for CCE haven't been discussed (unless I missed it) elsewhere in the manuscript.

This part of the discussion has been rewritten, and the language is gone.

L504 – Do you have any information on the immigrants, do they differ in their song?

We have identified three songs from the 1980s and 1990s that have at least two notes and/or structural features that differ from other Kent Island songs.

1982: no buzz, unusual low trill instead of high note cluster (no click train or high note cluster)

1996: double trill, short buzz (high note cluster)

1998: many unusual features; w-shaped note instead of click train / high note cluster.

Sonograms of these songs are presented in a new figure, Extended Data Figure 7, which is referred to in the new paragraph in the discussion on the possible origins of click trains (lines 306-308).

L633 – can you explain more about these potential precursors? This seems potentially very interesting! (Maybe I missed it in the main text, if so apologies).

Information on the possible origins of click trains can be found in the results section (lines 195-205).

L666 – what was the sung response? Did it differ in any way from the song that those males usually sung?

We did not record songs sung during the playback study (we did note how many songs were sung; an average of 0.6 per trial, which would not have provided a large enough sample size to work with). This has been noted in the Playback Methods (lines 717-719).

Figure captions 1 & 2 – these are very long, I would suggest revising to be more concise.

This has been done: 3 lines were cut from the Figure 1 caption, and 8 lines were cut from

the Figure 2 caption. The caption for Figure 1 is still rather long, but it is a large, complex figure.

Reviewer 3

Review of "Directional Selection and Cumulative Cultural Evolution in a Wild Songbird"

This study contributes to an expanding body of work on how social learning biases influence the cultural evolution of populations, and provides an alternative method of generative inference using discrete time modeling for simpler cultural systems (i.e. one variant replacing another, rather than a large number of variants [Lachlan et al., 2018, Nature Comms]). The model appears to be well-executed and incorporates important life-history information about the species. Additionally, this study is the first to follow up this kind of simulation-based work with playback experiments, to demonstrate that the new variant spreading through the population elicits greater responses from both females and males, and with assessments of the reproductive success of males singing particular variants. Overall, I think it represents an important advance in cultural evolution in non-human animals.

That being said, I have several critiques, two of which are major conceptual points.

First, the study is framed as the first "direct evidence of naturally occurring cumulative cultural evolution in a wild animal population". The results of this study meet the criteria for cumulative culture evolution (CCE) proposed by Mesoudi and Thornton (2018, Proc Roy Soc B), but according to those criteria, previous studies have already provided evidence for CCE, for instance in pigeons (Sasaki & Biro, 2017, Nature Comms) and zebra finches (Feher et al., 2009, Nature). In addition, these criteria appear to be too conservative, as they basically describe exactly what we would expect from cultural evolving traits that are sexually selected. This study demonstrates that social learning biases (in this case content-based bias, which the authors refer to as "cultural selection" [see below]) can drive the replacement of a less attractive cultural variant with a more attractive one, which does not seem to provide any new evidence for CCE. The majority of papers on the topic use a more strict definition for CCE that requires an increase in either complexity or efficiency (Dean et al., 2014, Biol Revs; Schofield et al., 2018, Primates), neither of which are demonstrated here. I feel that framing the study around cumulative cultural evolution is inaccurate and distracts from the core finding about content-based bias. My recommendation would be to remove all references to cumulative cultural evolution entirely.

We thank the reviewer for this comment, and we have rephrased our definition of cumulative cultural evolution (lines 68-71) to include an increase in efficiency as well as successive changes in socially learned behaviors. The increase in click number within click trains does meet this criterion – as Figure 4a shows, the increase in the number of clicks came at least 10 generations after the introduction of click trains, and our playback experiments show that longer click trains elicit stronger responses from both males and females, making the signal more effective.

Second, the authors do not present a clear notion of "cultural selection", which according to lines 92-94 appears to be content-based bias, or a preference for traits based on some aspect of their content. Throughout the paper, the authors make a dichotomy between "learning bias" and "selection", even though, in their own words, frequency-based bias is also a selective process (see lines 46-48). The more accurate framing would be frequency-based bias vs. content-based bias, as used by Lachlan et al. (2018, Nature Comms). Rendell et al. (2011, Trends Cogn Sci) have a great diagram clarifying the different forms of social learning biases

The reviewer's comment was very useful. Looking at Rendell's paper again helped in organizing terminology for the different types of mechanisms that might be responsible for the changes we observed. Box 3 was particularly helpful; we have used terminology based on the three categories described and depicted there as the basis for setting up the mechanisms in the introduction and then throughout the paper:

- Unbiased transmission, or drift (random copying)
- Frequency dependent bias/selection (conformist or rare-trait learning)
- Directly biased transmission or direct selection (which includes both content-dependent or sensory biases as well as model-based biases)

This framework is presented in lines 68-71.

Other points of criticism are:

- The first sentence of the abstract has several problems. Cultural evolution is the change in socially learned behavior, not the learning (transmission) of behavior itself. Only some learned behavior is socially learned and therefore relevant to culture. Unfortunately the authors' concept of culture does not specify social learning. This leads to further problems interpreting the rest of the sentence: what does it mean for learning to happen within and across lineages and generations? The subtext one must gather is that the learner is one individual and the one learned from is another individual, and these can be in the same or different lineages and generations, but none of this is clear from the sentence.

The abstract has been recast in response to this and other comments from reviewers (as well as the 150-word limit); mention of lineages is now gone.

- The last sentence of the abstract has the word driver twice. It appears that the authors mean that sexual selection is the major driver. Cumulative cultural evolution is not a driver of change, it is a description of the change itself.

The abstract, including the last sentence, has been completely rewritten and the language is gone.

- Why did innovation rate assume a single innovation rate, rather than including it as a parameter with a distribution informed by the cited studies? Also, varying the mutation rate and numbers of innovators would permit the researchers to see how much their assumptions shape the results.

We did run the model with a range of innovators, but did not initially include this in the paper out of concerns about length. The figure and table reporting this information have put into the Extended Data section (Table 4 and Figure 4), and the outcome reported in the results section in the same paragraph where we consider year of introduction (lines 178-182). Briefly, the number of innovators does make a difference to how well the model fits the data, with larger numbers of innovators increasing the fit. However, increasing the number of innovators had only very minor effects on the best fit values for conformity and selection. Given that the number of innovators did not affect the critical results, and because, as we point out in the Extended Data section, the methods, and in briefly in the results, larger numbers of innovators appear to be biologically unrealistic both in terms of our own observations and in others' data, we chose to use two innovators as our default within the model.

-The authors cite a previous paper (2013) from the same research group showing click trains replacing high note clusters. They actually cite this paper as the source of this result, despite claiming in their abstract that it was a major result of the present paper. This is not a minor point, because a main claim, even in the title of the paper, depends on this being novel. The present paper cannot claim that cumulative cultural evolution has never been documented if it was documented in this very population in 2013 (at the very least—in fact it has been documented elsewhere as well, and their mention of “cumulative” adds nothing to the interpretation of their results or to the paper’s novelty, as I mentioned above. The interesting thing about this study is really something else: its identification of the mechanism driving the change.)

It is true that the observations of change in the introductory segment of the song have been published previously. This new study presents two important findings that take it beyond the initial observation: first, the modeling of the potential roles of different cultural evolution / social learning mechanisms that might have driven the change to click trains, which allows us to demonstrate that frequency-biased learning was not important for this change, while direct selection alone can account for the changes we observed. Our analysis also allows us to demonstrate that cumulative cultural evolution did occur because a) clicks were not added to click trains until after a super-majority (> 75%) of songs included click trains, and b) longer click trains were more effective for eliciting responses from males and from females, making those longer click trains a more effective signal. We thank the reviewer for pointing out that we needed to frame the argument for cumulative cultural evolution more effectively, by emphasizing that two additional important criteria for cumulative cultural evolution have now been met: stepwise changes in a trait, and a result increase in efficiency. The wording of the abstract now reflects these points, as does the framing at the end of the introduction and the relevant parts of the discussion.

-The ending (last sentence) is hasty, like the beginning. There is no such thing as “horizontally across generations”.

The discussion has been rewritten, and the language has been deleted.

- One point that is highlighted by their research but not discussed is the important question of whether the meme is the thing with fitness or the individuals with the “meme”. Their fledging and sexual selection data suggest the latter. Cultural evolutionists have typically presumed the former. I wonder what the authors think.

We have rewritten the survival analysis in the results to better illustrate the point that adult males’ survival from year to year is not related to whether they sang click trains or high note clusters; rather, the replacement arose because young males preferentially copied click trains (lines 239-249). This finding is also highlighted in the first paragraph of the discussion.

In the absence of a biological fitness advantage (in the sense of natural selection), there is still a potential sexual selection advantage to singing click trains (and longer click trains). In fact, our data appear to say that such an advantage existed and that males singing click trains had a more salient signal in terms of responses by other males and by females.

Given the lability in song learning and the observation that survival from year to year did not differ for males singing click trains and high note clusters, we suggest in the results that some combination of demonstrator or payoff bias (observing that females are attracted

to specific males or song types) and sensory predispositions may be responsible for the direct selection we observed.

-The authors' claim that high note clusters are replaced by click trains is a seemingly concrete result that conceals some ambiguity. This issue is demonstrated in Figure 1. In song a, a high note cluster is essentially a click train followed by a high note and then a more rapid click train. Song b shows that the second click train can vary continuously into what could be called a trill. Regardless, saying that high note clusters were replaced by click trains might be a little too neat for what really happened, since high note clusters appear to include at least one click train. This terminology was set up in their 2013 paper, which has already passed review, but I believe it is important to get the wording and descriptions of their acoustic features clear and (if necessary) correct for this paper, because the distinction between these two forms is foundational for all of the results in the current paper. Here are some closeups of the spectrograms to make this issue clearer:

From figure 1, song a:

This is the "variable note" section of the high note cluster. It's followed by a high note, and then by:

This is a "trill" that ends the high note cluster.

This is a "click train", the completely different trait that replaced the sequence partially described above.

Here three different words (variable note, trill, click train) are being used to describe an acoustic signal that can be in fact very similar or even identical. The click train is said to have replaced the high note cluster in the population, but many high note clusters are comprised partly of one or more click trains. This was not clear in the manuscript, and actually influences the results. The initial "variable note" section could be just a click train, among other things. The "trill" section apparently varies continuously between a discrete click train and a more rapid sequence approaching a buzz. The only categorically distinct aspect of the high note cluster is the intervening "high note". The authors should deal with this ambiguity either by having a more quantitative way of distinguishing between what they consider to be two different acoustic forms, or else changing the wording and categorizing to reflect the complexity of what really happened to the song over the decades, or both.

The reviewer's point makes it clear that we need to more clearly define click trains, describe their relationship to high clusters more carefully, and provide supporting information for the delineation of note types.

Both forms of interstitial notes (high note clusters and click trains) are delimited by boundaries: louder introductory notes and, in some songs, the first middle note (all are loud notes; most interstitial notes are much softer). Both click trains and high note clusters include more than one note, but click trains include only one note type (clicks).

The individual components of the trills that are most often the final component of the high note cluster do indeed resemble clicks, but are shorter, sung at lower amplitude, and spaced much more closely together (the spacing in a trill is less than 3-4 ms, while the spacing of clicks is never less than 9 ms) – this information is now included in the legend of Extended Data Figure 1.

The components of a click train appear as part of one of 38 high note clusters recorded in 1982 (but not in any that were recorded in 1980). However, click trains clearly differ from high note clusters in that a) they do not include any of the other interstitial note types, and b) no notes other than clicks are present within the interval between introductory notes. In

terms of replacement, click trains initially appeared in addition to high note clusters, preceding them within the introductory sequence. The high note clusters were later deleted and thus “replacement” was a two-step process.

To clarify these points within the paper, we have:

- Added an explicit definition and an explanation of the difference between high note clusters and click trains early in the Results section (lines 101-104).
- Added a new figure (Extended Data, Figure 1) that provides definitions of note types and parametric support for the presence of distinct note types based on length of the notes.
- Added a phrase at the end of the Results paragraph on sources of innovation to clarify the point that the precursor to click trains seems likely to have been a high note cluster similar to that in Figure 4b-vi (see lines 201-205) that was “stuttered”, as in the song shown in Figure 4b-iv.
- Echoed the idea that improvisation or a copying error could have provided the initial click train innovation in the Discussion (see lines 315-320).

(It has not escaped our notice that high clusters may also have arisen through innovation, perhaps by altering an introductory note so that it is not frequency-modulated and then singing interstitial notes immediately before and after this new high note, but since we have no data prior to 1980, when the high cluster was the dominant form, we have chosen not to discuss this possibility.)

Reviewers' Comments:

Reviewer #1:

Remarks to the Author:

I commend the authors for their thorough revisions to this manuscript. I am now much more convinced that the study provides compelling evidence for CCE in natural populations and have a much clearer understanding of how the authors came to their conclusions concerning the mechanisms underpinning cultural change. I think this work will make an important contribution to the literature, but I do think that some further tweaks would be useful to make the manuscript easier to follow and thus maximise its impact. I am sorry for putting the authors through the ordeal of multiple rounds of revisions and would ask them to treat my comments below as suggestions only. As far as I am concerned, this is a really nice paper, but it could be made even better.

Main comments:

1) The changes to the introduction have helped to clarify the different potential mechanisms of cultural selection as well as the distinction between cultural evolution and cumulative cultural evolution. However, this has come at an important cost, because it is now less clear what questions the current study seeks to address: a reader could get to line 82 of the introduction without a clear sense of the outstanding gaps in our knowledge that the study seeks to fill. In particular, while the title and abstract frame the occurrence of CCE as the central question in the study, this question now comes across as a bit of an afterthought in the final line of the introduction. It is also notable that the topic of CCE isn't even introduced until the third paragraph of the introduction. If the authors want to frame this as a study about CCE (does it occur and what mechanisms drive it?) then I would suggest reordering the introduction. I think the introduction to animal culture and cultural evolution could be much shorter, allowing you to dive straight into CCE and to highlight the paucity of evidence in wild animals. This could then lead onto discussion of mechanisms underpinning cultural change (focusing specifically on mechanisms that might enable CCE). Here, I think the connection between the two themes ((1) CCE and (2) mechanisms of cultural change) could be clearer: how does the latter illuminate the former? Are certain mechanisms necessary for, or diagnostic of CCE?

2) I think it would also be helpful to end the introduction by outlining specific predictions: if the sparrows exhibit CCE (as opposed to cultural evolution more generally), what would we expect to see? Similarly, what would the predictions be for different mechanisms (drift vs frequency-dependence vs direct selection?). This would allow readers to put the results in context more easily as they go through the next section.

3) In the Discussion, I found the arguments concerning what is under selection rather confusing. You state on line 335 "that singing click trains – and, later, longer click trains – confers an advantage to males that is independent of their genetic "quality"; it is the signal rather than the male that is under selection." This seems to be based on a lack of clear survival benefits associated with song variants. However, if the signal is more "efficient" (i.e. elicits stronger responses from audiences) this could indeed provide a selective benefit to the singer. Indeed, you imply as much on line 266 when you say "the number of clicks in a train was important for both male competition and female choice". The outcomes of male competition and female choice are both likely to have important impacts on reproductive fitness (note that natural selection ultimately acts on lifetime reproductive success, not survival per se). It is also conceivable that the ability to learn or sing long click trains is related to an individual's genetic quality – at the moment there I see no evidence either for or against this possibility.

Minor comments:

L68-71: If you are following Mesoudi & Thornton's core criteria, I think it would be worth stating these explicitly, to help readers evaluate the evidence. At the moment the definition you give is rather loose

and does not specify the need for social learning.

L71: "Direct evidence comes from"... this sentence seems to equate the strength of evidence from experimental studies that clearly fulfil Mesoudi & Thornton's core criteria for CCE (pigeons and zebra finches) with much more speculative (and arguably unconvincing) evidence from Japanese macaques. The macaque example would fit better in the next sentence on "indirect or incomplete" evidence.

L330-343: Related to my main comment (3) above, I wonder how the arguments in this paragraph link to the previous evidence that males that sang click trains fledged more offspring than those that sang high note clusters. It feels a bit odd not to incorporate this evidence on differential reproductive success into the arguments here.

L374: What makes a socially learned behaviour "complex"? The word "complex" strikes me as intrinsically vague. I'd suggest either explaining what you mean by it or removing it.

Reviewer #3:

Remarks to the Author:

Review of revised Nature Communications NCOMMS-20-26259A-Z

"Direct Selection and Cumulative Cultural Evolution in Wild Bird Songs"

The authors have revised the manuscript rather extensively, such that most of the issues that were raised by myself (Reviewer 3) and Reviewer 2 are either rectified or rendered more evident such that the readers will be informed enough to make their own decisions.

The title says "direct selection", which is not a typical term in cultural evolution but is in genetic evolution. They cite Boyd & Richerson (1985) for this, who do not actually use that phrase, although they do use "direct bias" which appears to be what the authors are referring to. Today it is usually called "content bias". This is not a major issue, as the authors make clear what they are referring to, and direct or content bias is indeed a form of cultural selection.

The issue of cumulative cultural evolution still looms large, as finding it in bird song is a major claim of the paper. The revision is clear as to the meaning of this term. However, their hallmark of improved function is called "efficiency", whose meaning in the paper is vague but presumably has to do with how efficacious, or effective, the song is in terms of female choice. Elsewhere recently the lead author uses "efficiency" for such things as communicability, ease of learning, or physiological costs. However, on a common understanding of the term an effective song is not necessarily efficient one. Moreover the present study does not suggest that songs with click trains communicate better, are easier to produce or learn, or are better on any other measure of efficiency. Rather, apparently females like them better. Females liking them better makes them more effective, but does it make them more efficient? Moving beyond this to the bigger issue, the authors appear to be saying that if a song changes in a way that aligns better with female choice, its function is "improved" in the sense of the criterion for "cumulative cultural evolution". An issue with this is that there is nothing in the use of click trains that makes song objectively better or worse—it is only improved because females like it. Five decades ago females might not have liked them, in which case the authors would not have considered more click trains to demonstrate cumulative cultural evolution. The big question, then, is whether a "moving target" like female choice is sufficient to be the basis for "improvement" of a trait in relation to cumulative culture. Among writers on human cultural evolution the answer would probably be a resounding no. Science and technology are considered to be cumulatively culturally evolving because their results are objective improvements, i.e. improvements irrespective of what the interactants just happen to like. A fashion trend like a change in the wideness of popular ties, on the other hand, is not considered cumulative cultural evolution because the change in the width of ties is

not functional except in that people today like them a certain width. Thus the change being merely related to a preference of other humans disqualified it as counting as improvement. The present authors, however, allow for precisely this sort of fashion trend in bird song to be considered cumulative cultural evolution.

A related question but more empirical rather than semantic is: what is the difference between this study and any that claims to show response to sexual selection on any particular feature of bird song? Any study that shows a change in male song concordant with an increase in female preference for that form of song will apparently be viewed by these authors as demonstrating cumulative cultural evolution. But there are at least two alternative interpretations here. One is cultural selection, where males are choosing either what songs to learn or produce (what the authors apparently favor, calling it "direct selection"). The alternative is selection by females, as follows: (1) females started preferring click trains, (2) standing variation in male production of click trains was present, (3) females mated more with males who sang click trains, resulting in (4) a response to sexual selection, i.e. click trains spreading in the population, whether through their increased representation in the first generation (cultural evolution) or by any genetic predisposition to learn or produce click trains being sexually selected (genetic evolution). In other words, did the click trains spread because of a change in singing by males, i.e. a learning or production bias, or did females exert choice which winnowed standing variation? By the way, female choice is being assumed in this discussion but the question arguably remains the same if male competition is involved.

Part of the problem is no fault of the authors: this paper highlights the difficulty of making assumptions about the mechanism of birdsong change when this particular behavior is a poster child for two distinct modes of behavioral change. Bird song is socially learned, and so many are comfortable assuming that any change across generations is cultural evolution, i.e. manifested in the social learning process, such as through bias in the choice of particular song features, songs, or individuals to imitate. However, bird song is also a notable trait in studies of sexual selection, which can only be a mechanism of biological evolution insofar as it operates on heritable (rather than socially learned) features of song. And there are other possible mechanisms of change as well (that always get short shrift) but this distinction is sufficient to show the pickle the authors are in. Their tests look very much like tests of evolution by sexual selection, yet their conclusions are explicitly in terms of cultural evolution and they assume that mechanism.

The results of this study are wonderful and the paper deserves to be published in a prominent place. The questions and critique here are entirely about how the authors "sell" the results in terms of efficiency, cumulative cultural evolution, and to a lesser extent in their assumptions as to mechanism. If they cover these issues explicitly or else just avert them by, for instance, avoiding using the term cumulative culture altogether, the paper would likely be appropriate for the journal.

A minor empirical issue was raised by two reviewers but was not handled to the extent that the issue deserves. In playback experiments, two reviewers suggested comparing current response to click trains to response to high note clusters. The authors say this would conflate efficiency with familiarity, because the population no longer sings high note clusters. Perhaps the authors could mention the advantage that could be gained by testing naïve females on both sorts of preference. If only familiar stimuli are tested, we are getting only part of the story. Of course, this is a different study and requires different methods (e.g. handrearing).

Overview of changes in the April 2022 revision

Reviewer 1's suggestion for reorganizing the introduction and including Mesoudi and Thornton's [25] core criteria for cumulative cultural evolution as well as specific predictions were very valuable and we have followed those suggestions.

Mesoudi and Thornton's core criteria for cumulative cultural evolution are:

- 1) Generation of a variation of a behaviour
- 2) Social learning/transmission of the new variant form of the behaviour
- 3) Improvement in the "performance" (or "efficacy" or "efficiency") of the behaviour
- 4) Repetition of steps 1-3, producing incremental changes in the same behaviour.

We have laid these criteria out in paragraph 2 of the Introduction, explained that we had previously described one round of steps 1-3, and then stated that a second round of the same three steps producing further incremental cultural evolution within the same behaviour is necessary to provide evidence for cumulative cultural evolution.

We then, as Reviewer 1 suggested, go on to introduce the potential mechanisms responsible for the new forms of the behaviour spreading through the population, and provide specific predictions for each of the three main mechanisms.

After these changes had been made in the Introduction, bringing cumulative cultural evolution to the fore in paragraphs 2 with predictions in paragraph 3 and then introducing the mechanisms responsible for the changes in paragraph 4 and predictions in paragraph 5, it became clear that the Results and Discussion should consider these two topics in the same order as within the Introduction. Accordingly, we reorganized those sections to cover the evidence for cumulative cultural evolution first and then present the modeling results that provide evidence for a selection mechanism. This also required a re-ordering of the figures (and some of the extended data figures). The narrative flow works much better with the parallel structure across sections of the paper.

We have also edited our discussion of the "singer vs. song" point to make the background clearer and our argument more rigorous (see specific responses to reviewer 1).

Providing a better theoretical framework and specific criteria for cumulative cultural evolution also addresses issues underlying some of Reviewer 3's reservations about the use of the term and how well our data support it).

We have carefully considered all of Reviewer 3's comments. As we see it, this reviewer raised two related important conceptual issues (beyond that of the definition of cumulative cultural evolution, which we have addressed through our changes in response to Reviewer 1:

- 1) How sexual selection fits into the framework of mechanisms: direct selection, frequency-dependent biases, and drift. We have clarified this issue in the Introduction and Discussion (see detailed responses to Reviewer 3). Briefly, our formulation of direct selection follows that of Rendell et al. (2011), who used the term direct selection as we do, and Boyd and Richerson (1985), who used the term direct bias. Direct selection includes all forms of selection that favor social learning of a particular behaviour: prestige bias (e.g. copying dominant individuals), payoff bias (copying behaviours that succeed), sensory predispositions (whether learned or genetically based), and sexual selection (any bias towards copying a particular behavior because it is useful in male competition or female choice). The first two modes of direct selection are not directly analogous to forms of selection in organic evolution, but the last two are. Our data strongly suggest that direct sexual selection did indeed promote the cultural evolution of song features we described: reproductive success and playback studies point to greater efficacy of, first, songs with click trains, and later, songs with more clicks, in both the competition and choice modes of direct sexual selection. But sexual selection is not an alternative hypothesis, it is a subset of mechanisms subsumed by “direct selection”, which refers to cultural selection based on the behaviour itself rather than the frequency of a behaviour within the population, or lack of selection. We have amended the Introduction and Discussion (see detailed responses below) to clarify this point.
- 2) The reviewer asks whether changes in what can be described as “ornaments” (including songs) rather than “technological” (tools and machines) can be considered to be cumulative cultural evolution, suggesting that successive changes to behaviours that are ornaments cannot accumulate changes that increase overall function the same way as, say, changes to kayak construction may. As we point out in our responses below, changes to technology may also be driven by “fashion”, and so it can be hard to separate the two. Nevertheless, the reviewer is correct to point there may be an important difference between material artefacts such as tools and social artefacts such as pottery ornamentation, poetry, language, and music (all of which have been held to show cumulative cultural evolution) to which we add birdsongs. We have inserted the following sentence in the discussion to address this point:
“The cumulative cultural evolution we observed in Savannah sparrow songs is thus more akin to that of human social artefacts such as language⁹⁰, pottery ornamentation styles⁹¹ or music⁹² than to that of human material technology⁹³.”

Again, we thank both reviewers for their serious and careful consideration of the conceptual issues that are important in this paper. There is no doubt that the changes they have spurred have improved the presentation of our arguments and data.

Responses to Reviewer #1

I commend the authors for their thorough revisions to this manuscript. I am now much more convinced that the study provides compelling evidence for CCE in natural populations and have

a much clearer understanding of how the authors came to their conclusions concerning the mechanisms underpinning cultural change. I think this work will make an important contribution to the literature, but I do think that some further tweaks would be useful to make the manuscript easier to follow and thus maximise its impact. I am sorry for putting the authors through the ordeal of multiple rounds of revisions and would ask them to treat my comments below as suggestions only. As far as I am concerned, this is a really nice paper, but it could be made even better.

Both the reviewer's previous comments as well as those provided here have led us to make changes that have substantially improved the paper. We are grateful for their careful and thorough work.

Main comments:

1) The changes to the introduction have helped to clarify the different potential mechanisms of cultural selection as well as the distinction between cultural evolution and cumulative cultural evolution. However, this has come at an important cost, because it now less clear what questions the current study seeks to address: a reader could get to line 82 of the introduction without a clear sense of the outstanding gaps in our knowledge that the study seeks to fill. In particular, while the title and abstract frame the occurrence of CCE as the central question in the study, this question now comes across as a bit of an afterthought in the final line of the introduction. It is also notable that the topic of CCE isn't even introduced until the third paragraph of the introduction. If the authors want to frame this as a study about CCE (does it occur and what mechanisms drive it?) then I would suggest reordering the introduction. I think the introduction to animal culture and cultural evolution could be much shorter, allowing you to dive straight into CCE and to highlight the paucity of evidence in wild animals.

The opening paragraph has been substantially trimmed by summarizing main points and removing the lists of examples, cutting a total of 18 lines.

We then introduce cumulative cultural evolution (CCE) in the first words of the second paragraph, which presents the core criteria of Mesoudi and Thornton [25]. The next (third) paragraph explicitly describes how the new results reported in this paper satisfy these core criteria.

We note that one of these core criteria for CCE, that of improvement in the new form of the behavior, is often worded in the literature as "increased efficiency". The word "efficiency" is often confusing to those not familiar with the CCE literature (the animal communication literature uses the word efficiency to describe a different concept). Thus, we have used the word "efficacy" as well as the phrase "improved performance", both of which have meanings more appropriate to our measurements of the effects of cultural evolution.

This could then lead onto discussion of mechanisms underpinning cultural change (focusing specifically on mechanisms that might enable CCE). Here, I think the connection between the

two themes ((1) CCE and (2) mechanisms of cultural change) could be clearer: how does the latter illuminate the former? Are certain mechanisms necessary for, or diagnostic of CCE?

The fourth paragraph now introduces mechanisms for cultural evolution generally (rather than CCE in particular). The fifth paragraph then describes how we used modelling to make specific predictions, and suggests that cultural selection mechanisms may differ in different rounds (core criteria i-iii) of CCE.

2) I think it would also be helpful to end the introduction by outlining specific predictions: if the sparrows exhibit CCE (as opposed to cultural evolution more generally), what would we expect to see? Similarly, what would the predictions be for different mechanisms (drift vs frequency-dependence vs direct selection?). This would allow readers to put the results in context more easily as they go through the next section.

This is an excellent suggestion. It seemed to make more sense to insert the predictions in context within the Introduction: in the third paragraph we provide criteria for demonstrating CCE (based on Mesoudi and Thornton's core criteria), and in the fifth paragraph we provide predictions and hypotheses related to mechanisms of cultural evolution and for CCE more generally. We consider the explicit statements of these predictions and hypotheses to be very helpful additions.

3) In the Discussion, I found the arguments concerning what is under selection rather confusing. You state on line 335 "that singing click trains – and, later, longer click trains – confers an advantage to males that is independent of their genetic "quality"; it is the signal rather than the male that is under selection." This seems to be based on a lack of clear survival benefits associated with song variants. However, if the signal is more "efficient" (i.e. elicits stronger responses from audiences) this could indeed provide a selective benefit to the singer. Indeed, you imply as much on line 266 when you say "the number of clicks in a train was important for both male competition and female choice". The outcomes of male competition and female choice are both likely to have important impacts on reproductive fitness (note that natural selection ultimately acts on lifetime reproductive success, not survival per se). It is also conceivable that the ability to learn or sing long click trains is related to an individual's genetic quality – at the moment there I see no evidence either for or against this possibility.

We have now set up this part of the Discussion by introducing the selection on the song vs. selection on the singer question in the revised Introduction (lines 86-87). Our point is that it is the song that a male has learned to sing, not the intrinsic "genetic quality" of the male, that confers the reproductive advantage to that male, and that reproductive fitness (the number of offspring a male produces) is not necessarily coupled to cultural fitness (the number of times a male's song is copied). We have also clarified the language about this point in the Discussion (see lines 354-362).

Minor comments:

L68-71: If you are following Mesoudi & Thornton's core criteria, I think it would be worth stating these explicitly, to help readers evaluate the evidence. At the moment the definition you give is rather loose and does not specify the need for social learning.

An excellent suggestion, and it has been implemented in the second and third paragraphs of the Introduction.

L71: "Direct evidence comes from"... this sentence seems to equate the strength of evidence from experimental studies that clearly fulfil Mesoudi & Thornton's core criteria for CCE (pigeons and zebra finches) with much more speculative (and arguably unconvincing) evidence from Japanese macaques. The macaque example would fit better in the next sentence on "indirect or incomplete" evidence.

As suggested, the Japanese macaque citation has been moved to the "indirect or incomplete evidence" sentence (lines 45-6).

L330-343: Related to my main comment (3) above, I wonder how the arguments in this paragraph link to the previous evidence that males that sang click trains fledged more offspring than those that sang high note clusters. It feels a bit odd not to incorporate this evidence on differential reproductive success into the arguments here.

To address this point, the evidence for differential reproductive success of males singing click trains as opposed to high note clusters is now cited in lines 49-51 of the Introduction, and is also covered in the Discussion (lines 339-41).

L374: What makes a socially learned behaviour "complex"? The word "complex" strikes me as intrinsically vague. I'd suggest either explaining what you mean by it or removing it.

As suggested, the word "complex" has been removed from the final sentence of the Discussion (line 378).

Responses to Reviewer #3

The authors have revised the manuscript rather extensively, such that most of the issues that were raised by myself (Reviewer 3) and Reviewer 2 are either rectified or rendered more evident such that the readers will be informed enough to make their own decisions.

We thank reviewer 3 for raising the important points that urged us to address the issues of material vs. social artefacts (tools vs. ornaments) and where sexual selection fits within our conception of mechanisms. We have clarified our presentation of the conceptual framework and have carefully discussed our results within that framework. See below for details of the changes.

The title says “direct selection”, which is not a typical term in cultural evolution but is in genetic evolution. They cite Boyd & Richerson (1985) for this, who do not actually use that phrase, although they do use “direct bias” which appears to be what the authors are referring to. Today it is usually called “content bias”. This is not a major issue, as the authors make clear what they are referring to, and direct or content bias is indeed a form of cultural selection.

The wording on line 71 has been adjusted to make it clear that we are using the term “direct selection”, selection based on the properties of the trait, is used in contrast to “frequency-based selection” which is based on the prevalence of the trait. We have added a reference to Rendell et al.’s 2011 review [55], which lays this out nicely, to the sentence in question (lines 79-81). Direct selection is not exactly the same as content bias, as it includes prestige bias as well as sensory predispositions and payoff bias.

The issue of cumulative cultural evolution still looms large, as finding it in bird song is a major claim of the paper. The revision is clear as to the meaning of this term. However, their hallmark of improved function is called “efficiency”, whose meaning in the paper is vague but presumably has to do with how efficacious, or effective, the song is in terms of female choice. Elsewhere recently the lead author uses “efficiency” for such things as communicability, ease of learning, or physiological costs. However, on a common understanding of the term an effective song is not necessarily efficient one. Moreover the present study does not suggest that songs with click trains communicate better, are easier to produce or learn, or are better on any other measure of efficiency. Rather, apparently females like them better. Females liking them better makes them more effective, but does it make them more efficient?

The reviewer is right. The term “efficiency” (for improvement in the behavior’s outcome or learnability) as used in the CCE literature (see titles of references [26] and [44], can be confusing, especially as the animal communication literature uses the word efficiency to describe a different concept. As mentioned above, we have substituted the word “efficacy” as well as the phrase “improved performance”, which have meanings closer to our measurements for improvements in the behavior as specified in the third core criterion for CCE.

Moving beyond this to the bigger issue, the authors appear to be saying that if a song changes in a way that aligns better with female choice, its function is “improved” in the sense of the criterion for “cumulative cultural evolution”. An issue with this is that there is nothing in the use of click trains that makes song objectively better or worse—it is only improved because females like it. Five decades ago females might not have liked them, in which case the authors would not have considered more click trains to demonstrate cumulative cultural evolution. The big question, then, is whether a “moving target” like female choice is sufficient to be the basis for “improvement” of a trait in relation to cumulative culture. Among writers on human cultural evolution the answer would probably be a resounding no. Science and technology are considered to be cumulatively culturally evolving because their results are objective improvements, i.e. improvements irrespective of what the interactants just happen to like. A fashion trend like a change in the wideness of popular ties, on the other hand, is not considered cumulative cultural evolution because the change in the width of ties is not functional except in that people today like them a certain width. Thus the change being merely related to a preference of other humans disqualified it as counting as improvement. The present authors, however, allow for precisely this sort of fashion trend in bird song to be considered cumulative cultural evolution.

The issue of what constitutes an improvement of performance can become confusing when considering different types of cultural phenomena. In the case of a complicated tool that is the product of human cultural evolution, it would seem that improvement in performance is different from changes in ornamentation, as in the reviewer’s tie width example. But ornamentation has a function, to elicit responses in the viewer (or listener), and the literature on the cultural evolution and the cumulative cultural evolution of ornamentation in human cultures is extensive (see Shennan’s extensive work on pottery styles, some of which is summarized in reference 80). Even if we consider technological innovations, fashion may drive cultural change. If it becomes more fashionable or convenient for members of a population to use natural gas to heat their homes, then innovations in boilers that improve their performance with a different fuel will be favored by that population – but not because natural gas is inherently better than fuel oil for heating homes in all situations, or because natural gas boilers are inherently better than fuel oil boilers. Thus, what makes a change in an artefact or tool more effective can be due to changes in the preferences of the users. Likewise, consider the use of language: its evolution is a classic example of cumulative cultural evolution, yet specific words and phrases evolve not necessarily because they function better in a utilitarian sense, but because fashions in speech change (and these changes accumulate).

Research on cumulative cultural evolution in human language, art and music is well represented in the literature, and Savannah sparrow song CCE is more similar to these examples than to human material culture. We have added a sentence and references to the Discussion (see lines 351-3) to make this point.

A related question but more empirical rather than semantic is: what is the difference between this study and any that claims to show response to sexual selection on any particular feature of bird song? Any study that shows a change in male song concordant with an increase in female preference for that form of song will apparently be viewed by these authors as demonstrating cumulative cultural evolution. But there are at least two alternative interpretations here. One is cultural selection, where males are choosing either what songs to learn or produce (what the authors apparently favor, calling it “direct selection”). The alternative is selection by females, as follows: (1) females started preferring click trains, (2) standing variation in male production of click trains was present, (3) females mated more with males who sang click trains, resulting in (4) a response to sexual selection, i.e. click trains spreading in the population, whether through their increased representation in the first generation (cultural evolution) or by any genetic predisposition to learn or produce click trains being sexually selected (genetic evolution). In other words, did the click trains spread because of a change in singing by males, i.e. a learning or production bias, or did females exert choice which winnowed standing variation? By the way, female choice is being assumed in this discussion but the question arguably remains the same if male competition is involved.

(The points the reviewer raises in the paragraph above are related to those in the paragraph below, so we have provided one unified response, immediately after the next paragraph.)

Part of the problem is no fault of the authors: this paper highlights the difficulty of making assumptions about the mechanism of birdsong change when this particular behavior is a poster child for two distinct modes of behavioral change. Bird song is socially learned, and so many are comfortable assuming that any change across generations is cultural evolution, i.e. manifested in the social learning process, such as through bias in the choice of particular song features, songs, or individuals to imitate. However, bird song is also a notable trait in studies of sexual selection, which can only be a mechanism of biological evolution insofar as it operates on heritable (rather than socially learned) features of song. And there are other possible mechanisms of change as well (that always get short shrift) but this distinction is sufficient to show the pickle the authors are in. Their tests look very much like tests of evolution by sexual selection, yet their conclusions are explicitly in terms of cultural evolution and they assume that mechanism.

The reviewer raises an important point. Just as sexual selection can contribute to organic evolution, it also can contribute to cultural evolution – and in the case of bird song, which is used for male-male interactions and female choice, it clearly must do so. We did not make the point that direct selection in this case appears to be direct sexual selection sufficiently explicit in the previous version of the manuscript. We have now made this point much more explicit (see line 20 in the Abstract, lines 73-5 and 83-6 in the Introduction, and lines 341-350 the Discussion).

We do consider cultural sexual selection to be operating here, and we also consider sexual selection to be a component of direct selection in shaping song learning (as opposed to drift

and frequency-based selection). As we point out, our playback data suggest that both male competition and female choice may be playing a role (see also our response to the next comment. Just as both natural selection and sexual selection play a role in organic evolution, these two forms of selection can play a role in cultural evolution within the category of direct selection (as opposed to drift and frequency-based selection). If female preferences shape the social learning of songs by males, then sexual selection on the trait (which is a form of direct selection) results in cultural evolution, because the song is socially learned, and not transmitted genetically (as would be necessary for organic evolution). Our changes in response to reviewer 1's suggestions should clarify the argument as we now state it throughout the paper:

- Social learning that results in improved efficacy makes population changes in the trait a form of cultural selection;
- Two distinct rounds of changes in the same trait under cultural selection meet the core criteria for cumulative evolution;
- Our modeling, reproductive success measures, and playback study implicate direct selection, and specifically direct sexual selection, as the most important mechanism driving the changes in the socially learned trait.

Finally, the reviewer also suggests that sexual selection could have operated on standing variation in the production of click trains, winnowing that standing variation. Our data explicitly speak against this hypothesis. First, there were no click trains in 1980 or 1982, so there was no standing variation for the first round of cultural evolution. Second, for the next round of cultural evolution, young often birds sang more clicks than were present in the songs of older birds, increasing standing variation rather than winnowing it as would be the case in organic evolution. We have reworded the sentence in lines 332-5 of the Discussion to explain this point more clearly.

The results of this study are wonderful and the paper deserves to be published in a prominent place. The questions and critique here are entirely about how the authors "sell" the results in terms of efficiency, cumulative cultural evolution, and to a lesser extent in their assumptions as to mechanism. If they cover these issues explicitly or else just avert them by, for instance, avoiding using the term cumulative culture altogether, the paper would likely be appropriate for the journal.

With the inclusion, at the recommendation of reviewer 1, of an explicit listing of the core criteria for cumulative cultural evolution and an explanation of how our observations satisfy those criteria, the case for using CCE is made more explicit and hence stronger. These changes (in paragraphs 2 and 3 of the Introduction, the re-ordering of the Results, and the largely rewritten first paragraph of the Discussion, starting on line 261) have made the argument for CCE more explicit and strengthened it – and should satisfy reviewer 3's concerns about demonstrating CCE.

A minor empirical issue was raised by two reviewers but was not handled to the extent that the issue deserves. In playback experiments, two reviewers suggested comparing current response to click trains to response to high note clusters. The authors say this would conflate efficiency with familiarity, because the population no longer sings high note clusters. Perhaps the authors could mention the advantage that could be gained by testing naïve females on both sorts of preference. If only familiar stimuli are tested, we are getting only part of the story. Of course, this is a different study and requires different methods (e.g. handrearing).

A sentence covering this point has been inserted into playback portion of the Methods section (lines 619-22). As the reviewer notes, using hand-reared birds as subjects addresses a somewhat different question, given that the experiences of birds in natural populations may reflect social learning of preferences as well as stimulus salience (as we now mention in the Discussion; see lines 347-9).

Reviewers' Comments:

Reviewer #3:

Remarks to the Author:

The authors have responded to Reviewer 3's comments in helpful ways, both in the response and, more importantly, in the manuscript text.

All points have been dealt with adequately, except arguably for one: the nature of cumulative cultural evolution. However, with a couple of exceptions the manuscript now presents a position that would be acceptable to many other reviewers and readers in the field— thus, it is not sufficiently clear that the authors are in error, for a revision of this point to be required (i.e, for them to be required to take a position they do not hold). CCE is still problematic in the field of cultural evolution itself, with a recurring danger of collapsing into cultural evolution of any sort. The authors run this risk themselves. Thus, in case it is of value, here is a brief statement of the disputed perspective of the authors, and a specific example of a problematic statement in the text that might serve to highlight the issue.

The authors do not make a distinction between cultural evolution that improves access to fixed, external targets (e.g. migration routes and foraging tools), and cultural evolution that improves nothing more than the match of a display to a preference. If we consider the latter to be an example of cumulative cultural evolution (CCE), then all cultural evolution of bird song features in line with sexual selection is CCE. The authors do not say this, but it is the logical conclusion of their take.

The fuzzy division between CCE and non-CCE can be seen, for instance, in line 351 of the revised text: "The cumulative cultural evolution we observed in Savannah sparrow songs is thus more akin to that of human social artefacts such as language, pottery ornamentation styles or music than to that of human material technology."

Precisely! And the tradition in the field of cultural evolution is to consider the former three examples *not* to be CCE, because there is no clear standard by which languages, pottery ornamentation, and music are improving over time. In fact pottery ornamentation is routinely used in basic textbooks as an example of cultural evolution that is not cumulative in the functional sense of CCE.

We'll leave it at that— this represents a common issue with CCE in the broader literature, despite clarity of the definition, and so it is just represented here again in case the authors may benefit from it.

Minor suggestions:

The authors should take care not to too strongly separate out sexual selection from payoff biases and prestige biases which can themselves be sexually selected biases.

The first sentence of the Discussion is incorrect from the authors' own perspective. Perhaps they meant to add the word "cumulative".

Suggest changing "genetic trait" in line 334 to "heritable trait" or some such thing— all traits are underlain by genes.

Reviewer #4:

None

Reviewer #3 (Remarks to the Author):

The authors have responded to Reviewer 3's comments in helpful ways, both in the response and, more importantly, in the manuscript text.

All points have been dealt with adequately, except arguably for one: the nature of cumulative cultural evolution. However, with a couple of exceptions the manuscript now presents a position that would be acceptable to many other reviewers and readers in the field— thus, it is not sufficiently clear that the authors are in error, for a revision of this point to be required (i.e, for them to be required to take a position they do not hold). CCE is still problematic in the field of cultural evolution itself, with a recurring danger of collapsing into cultural evolution of any sort. The authors run this risk themselves. Thus, in case it is of value, here is a brief statement of the disputed perspective of the authors, and a specific example of a problematic statement in the text that might serve to highlight the issue.

The authors do not make a distinction between cultural evolution that improves access to fixed, external targets (e.g. migration routes and foraging tools), and cultural evolution that improves nothing more than the match of a display to a preference. If we consider the latter to be an example of cumulative cultural evolution (CCE), then all cultural evolution of bird song features in line with sexual selection is CCE. The authors do not say this, but it is the logical conclusion of their take.

The fuzzy division between CCE and non-CCE can be seen, for instance, in line 351 of the revised text: "The cumulative cultural evolution we observed in Savannah sparrow songs is thus more akin to that of human social artefacts such as language, pottery ornamentation styles or music than to that of human material technology."

Precisely! And the tradition in the field of cultural evolution is to consider the former three examples **not** to be CCE, because there is no clear standard by which languages, pottery ornamentation, and music are improving over time. In fact pottery ornamentation is routinely used in basic textbooks as an example of cultural evolution that is not cumulative in the functional sense of CCE.

We'll leave it at that— this represents a common issue with CCE in the broader literature, despite clarity of the definition, and so it is just represented here again in case the authors may benefit from it.

We thank the reviewer for this clarification of their perspective on CCE, and we agree the point is disputed in the broader literature. For example, Shennan and Wilkinson (2001) write:

"Following on the work of Dunnell, the evolutionary archaeology school has made a sharp distinction between functional and stylistic variation in archaeological artifacts. Variation is defined as functional if it is affected by selection processes and as stylistic if it is a result of processes of random drift."

We have explicitly shown here that the changes in the songs we describe are not due to random drift, so by this definition the song changes we describe are in fact functional and

so can undergo cumulative cultural evolution. That the function is based on preferences that may be based either in sensory predispositions or be learned certainly adds interest to the overall question. Nevertheless, what we have demonstrated clearly fits the standard definition of cumulative cultural evolution. (as described in Mesoudi and Thornton and explained the second paragraph of in the introduction). To address this issue, we have added three sentences to the discussion at line 356 (immediately after the sentence cited above by the reviewer), as follows, citing Shennan and Wilkinson:

In these realms, the distinction between “functional” and “stylistic” changes is often tied to mechanisms: stylistic changes are due to drift, while functional changes are due to selection⁸³. We have shown that drift cannot account for the changes in Savannah sparrow song introduction, which are due to selection, specifically sexual selection. That this selection is due to preferences that may be based on sensory predispositions or may themselves be learned (and evolve) adds interest to the scenario.

Minor suggestions:

The authors should take care not to too strongly separate out sexual selection from payoff biases and prestige biases which can themselves be sexually selected biases.

This is a good point. Biases may operate within natural selection and within sexual selection. We checked all the language around our presentation of prestige and payoff biases. We checked our language around this issue. We initially present all of these terms briefly in the introduction, (lines 69-74), and we do not relate biases and sexual selection there. However, in lines 351-353 we say:

It is likely that some combination of 1) demonstrator or payoff bias and 2) female sensory predispositions^{2,78} (which may themselves be learned) is responsible for direct sexual selection...

This language explicitly relates sexual selection to prestige and payoff biases, and points out that the two may operate together. We changed “demonstrator” to “prestige” to align the language better with the introduction.

The first sentence of the Discussion is incorrect from the authors’ own perspective. Perhaps they meant to add the word “cumulative”.

The reviewer may have misread this paragraph. The first two sentences refer to simple cultural evolution; the next two sentences make a clear transition to cumulative cultural evolution.

Suggest changing “genetic trait” in line 334 to “heritable trait” or some such thing— all traits are underlain by genes.

Changed to “heritable trait”.